# Latching dynamics as a basis for short-term recall

**Kwang Il Ryom**[1,©,*], **Vezha Boboeva**[1,©,¤], **Oleksandra Soldatkina**[1],
**Alessandro Treves**[1,2,*]

**1** Sector of Cognitive Neuroscience, SISSA, Trieste, Italy, **2** Kavli Institute for Systems Neuroscience/Centre for Neural Computation, NTNU, Trondheim, Norway

© These authors contributed equally to this work.
¤ Current address: Department of Bioengineering, Imperial College, London, United Kingdom
* kryom@sissa.it (KIR); ale@sissa.it (AT)

## Abstract

We discuss simple models for the transient storage in short-term memory of cortical patterns of activity, all based on the notion that their recall exploits the natural tendency of the cortex to hop from state to state—latching dynamics. We show that in one such model, and in simple spatial memory tasks we have given to human subjects, short-term memory can be limited to similar low capacity by interference effects, in tasks terminated by errors, and can exhibit similar sublinear scaling, when errors are overlooked. The same mechanism can drive serial recall if combined with weak order-encoding plasticity. Finally, even when storing randomly correlated patterns of activity the network demonstrates correlation-driven latching waves, which are reflected at the outer extremes of pattern space.

**Data Availability Statement:** Simulation code and experimental data are available from https://osf.io/ng2fw.

**Funding:** Work supported by Human Frontier Science Program Grant RGP0057/2016 (AT) and by EU Marie Skłodowska-Curie Training Network

## Author summary

What makes short-term memory so poor, that over a minute we tend to forget even phone numbers, if we cannot rehearse or record them electronically? In comparison, long-term memory can be amazingly rich and accurate. Was it so difficult to equip our brain with a short-term memory device of reasonable capacity?

We discuss the hypothesis that instead of an *ad hoc* device, short-term memory relies on long-term representations, and that the short-term recall of multiple items exploits the natural tendency of the cortex to jump from state to state, by only adding imprecisely determined "kicks" that spur cortical dynamics towards the states representing those items. We show that a plausible neural model for such kicks performs similarly to human subjects we have tested, both in conditions when short-term recall is terminated by errors, and when errors are overlooked and subjects are asked to keep trying. The same mechanism can drive serial recall, if combined with equally imprecise kicks encoding item order. Our analysis suggests that a proper short-term memory device may have never evolved in our brain, which had, therefore, to make do with tweaking its superb long-term memory capabilities.

765549 M-Gate (AT, OS). The funders had no role in study design, data collection and analysis, decision to publish, or preparation of the manuscript.

## Introduction

Despite much effort directed towards understanding the neural processes underlying short-term memory (STM), what causes its notoriously limited capacity has, to this day, remained largely mysterious [1–5]. If one were to take a functionalist perspective, inspired e.g. by Baddeley's theory of working memory [6], and assume that items in short-term memory are transiently represented in a dedicated cortical module, where they have been copied from their long-term traces, two riddles would arise: how would the copying work? and why would this module have such poor capacity? Multiple lines of evidence, particularly since the advent of functional imaging, have however failed to identify an *ad hoc* STM module, and indicated that STM is expressed by the activity of the same neurons that participate in the representation of long-term memories (LTM) [7]. This disposes of the copy riddle, but emphasizes the capacity one. What makes us able, for example, to recognize tens of thousands of images as familiar [8] and yet unable to detect a change in a configuration of more than a few elements that we have just seen [9]? Focusing on the recall of sequences of well-known items, what makes it so difficult to go, again, beyond very short sequences?

Addressing this riddle with a mathematically well-defined neural network model requires, in our view, a model that, however drastically simplified, captures the widely distributed nature of the cortical representations which STM as well as LTM can rely on. We argue that a Potts network is adequate in this respect [10]. A Potts network can model the long-range interactions among patches of cortex and, without any *ad hoc* component, shows a tendency to hop spontaneously from activity pattern to activity pattern, recalling them in a sequence resembling a random walk. We call this *latching dynamics* and propose here that it holds the key to understand STM limitations, once combined with some mechanism, perforce imprecise, for short-term storage. We consider a number of distinct mechanisms of this type, that by adding an extra "kick" to boost a small subset of $L$ among $p$ patterns in long-term memory, approximately restrict latching dynamics to the subset, which is then effectively kept in short-term memory.

We show that this formulation fits with the general hypothesis that interference between memories is critical [11] as well as with the gist of the recently proposed statistical theory of free recall, as implemented by stochastic trajectories among ensembles of items [12], in fact unifying them: depending on the task, the limiting factor turns out to be either interference from items in long-term memory or the randomness in retrieval trajectories.

While the basic model needs more structure to be predictive about specific behaviour, e.g. in semantic priming experiments [13], or about the effects of item complexity [14] or individual differences [15], and in general to fully benchmark its validity as a model of short-term memory [4], we show that it is consistent with simple experiments, that illustrate the way STM limitations depend on task demands. In free recall, where repetitions and mistakes are not penalised, the number $M$ of retrieved items tends to scale sublinearly with $L$, reflecting largely random exploration. In a task which is terminated by mistakes, instead, capacity is constrained by the interference of other items in long-term memory. Further, modeling *serial* recall with hetero-associative short-term synaptic enhancement leads to the conclusion that latching dynamics is preserved only if the enhancement is weak, and then it generates limited sequences, similar to those shown by human subjects when asked to serially recall unstructured items, without recourse to LTM aids.

The paper is organized as follows. In Models we first review the basics of the Potts network for LTM, and its tendency to latch; those familiar with it may go directly to the next subsection, which compares three ways to harness it for short-term memory, showing how they enable to analyze STM limits. In Results we look at the performance of the second mechanism in free

recall and compare it with experimental data, before modelling serial recall with the same model. The nature of the trajectories the model follows in memory space is further analysed in the last part of Results, with concluding remarks in Discussion.

## Models

### The Potts network for the storage of long-term memories

A Potts neural network is an autoassociative memory network comprised of $N$ Potts units, each of which represents the state of a single patch of the cortex as it contributes to retrieve distributed LTM traces addressed by their contents [16]. Each Potts unit has $S$ active states, indexed as $1, 2, \cdots, S$, representing local attractors in that patch, and one background-firing state (no local attractor is activated), the 0 state. The $N$ units interact with each other via tensor connections, that represent associative long-range interactions through axons that travel through the white matter [17], while local, within-gray-matter interactions are assumed to be governed by attractor dynamics in each patch (Fig 1A). The values of the tensor components

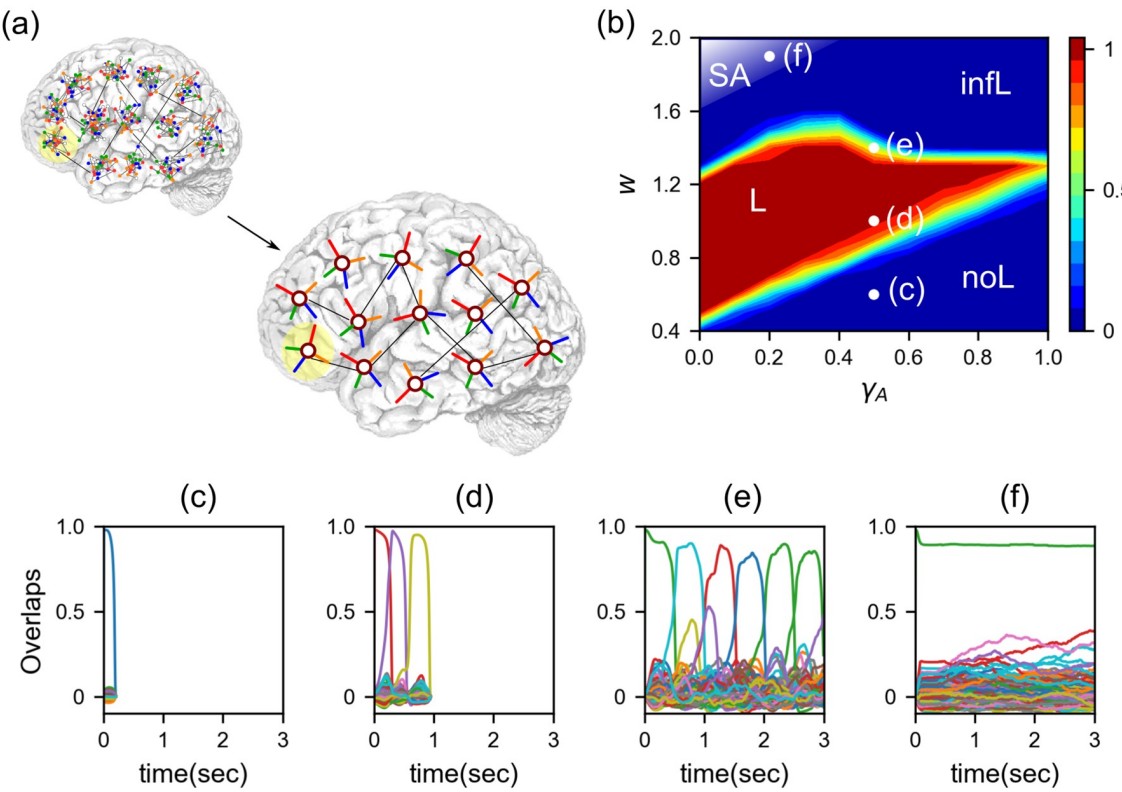

**Fig 1. Latching dynamics of the Potts neural network.** (a): The Potts network encapsulates local attractor dynamics within cortical patches into Potts spins and describes attractor dynamics in the global network of the cortex by means of a network of Potts units. (b): Phase diagram of a Potts neural network in $w - \gamma_A$ plane. The $x$-axis is $\gamma_A$, the proportion of fast inhibition in the dynamics. $\gamma_A = 0$ (1) means only slow (fast) inhibition. The $y$-axis is the self-reinforcement parameter $w$. In false color, the proportion of simulations that exhibit finite latching. Increasing $w$, in fact, one observes different latching phases: no latching (*noL*), finite latching (*L*), infinite latching (*infL*) and stable attractor phase (*SA*). White circles indicate four points, where examples of latching sequences are shown in the bottom panels, all produced with time constants $\tau_1 = 0.01s$, $\tau_2 = 0.2s$ and $\tau_3 = 100s$. The $x$-axis corresponds to time, and the $y$-axis to the overlap, each colour with an item in long-term memory. (c): For too low $w$, in the no latching phase, there is only retrieval and the network cannot latch onto another pattern. (d): Increasing $w$, one reaches the finite latching phase, where the network retrieves a finite sequence of patterns, with high overlap. (e): Increasing $w$ further, one reaches the infinite latching phase, where sequences are indefinitely long but the quality of latching is degraded. The mean dwell time in a pattern is also increased compared with the finite latching regime. (f): Increasing $w$ even further, one gets to the stable attractor phase, where the network retrieves the cued pattern and cannot escape from that attractor.

are pre-determined by the Hebbian learning rule, which can be contrued as derived from Hebbian plasticity at the synaptic level [10]

$$J_{ij}^{kl} = \frac{c_{ij}}{c_m a\left(1 - \frac{a}{S}\right)} \sum_{\mu=1}^{p} \left(\delta_{\xi_i^\mu k} - \frac{a}{S}\right)\left(\delta_{\xi_j^\mu l} - \frac{a}{S}\right)(1 - \delta_{k0})(1 - \delta_{l0}),$$ (1)

where $c_{ij}$ is either 1 if unit $j$ gives input to unit $i$ or 0 otherwise, allowing for asymmetric connections between units, and the $\delta$'s are the Kronecker symbols. The number of input connections per unit is $c_m$. The $p$ distributed activity patterns which represent LTM items are assigned, in the simplest model, as composition of local attractor states $\{\xi_i^\mu\}$ ($i = 1, 2, \cdots, N$ and $\mu = 1, 2, \cdots, p$). The variable $\xi_i^\mu$ indicates the state of unit $i$ in pattern $\mu$ and is randomly sampled, independently on the unit index $i$ and the pattern index $\mu$, from $\{0, 1, 2, \cdots, S\}$ with probability

$$P(\xi_i^\mu = k) = \frac{a}{S}(1 - \delta_{k,0}) + (1 - a)\delta_{k,0}.$$ (2)

Constructed in this way, patterns are randomly correlated with each other. We use these randomly correlated memory patterns $\{\xi_i^\mu\}_{\mu=1,\dots,p}$ in this study, but envisage later generalizing it to a set of correlated memory patterns, as produced by the algorithm presented in [18]. The parameter $a$ is the sparsity of patterns—fraction of active units in each pattern; the average number of active units in any pattern $\mu$ is therefore given by $Na$.

Local network dynamics within a patch are taken to be driven by the input that the unit $i$ in state $k$ receives

$$h_i^k(t) = \sum_{j \neq i}^{N} \sum_{l=1}^{S} J_{ij}^{kl} \sigma_j^l(t) + w\left[\sigma_i^k(t) - \frac{1}{S}\sum_{l=1}^{S}\sigma_i^l(t)\right],$$ (3)

where the local feedback $w$, introduced in [19], models the depth of attractors in a patch, as shown in [10]—it helps the corresponding Potts unit converge to its most active state. The activation along each state for a given Potts unit is updated with a *soft max* rule

$$\sigma_i^k(t) = \frac{\exp[\beta r_i^k(t)]}{\sum_{k=1}^{S}\exp[\beta r_i^k(t)] + \exp\{\beta[U + \theta_i^A(t) + \theta_i^B(t)]\}} \quad \text{if } k > 0,$$

$$\sigma_i^0(t) = \frac{\exp\{\beta[U + \theta_i^A(t) + \theta_i^B(t)]\}}{\sum_{k=1}^{S}\exp[\beta r_i^k(t)] + \exp\{\beta[U + \theta_i^A(t) + \theta_i^B(t)]\}} \quad \text{if } k = 0,$$ (4)

where $U$ is a fixed threshold common for all units and $\beta$ measures the level of noise in the system. Note that $\sigma_i^k$ takes continuous values in (0, 1) and that $\sum_{k=0}^{S}\sigma_i^k = 1$ for any $i$. The variables $r_i^k, \theta_i^A$ and $\theta_i^B$ parametrize, respectively, the state-specific potential, fast inhibition and slow inhibition in patch $i$. The state-specific potential $r_i^k$ integrates the input $h_i^k$ by

$$\tau_1 \frac{dr_i^k(t)}{dt} = h_i^k(t) - \theta_i^k(t) - r_i^k(t),$$ (5)

where the variable $\theta_i^k$ is a specific threshold for unit $i$ and for state $k$. If it were constant in time, the Potts network would simply operate as an autoassociative memory with extensive storage capacity [20].

Taking the threshold $\theta_i^k$ to vary in time to model adaptation, i.e. synaptic or neural fatigue selectively affecting the neurons active in state $k$, and not all neurons subsumed by Potts unit $i$

$$\tau_2 \frac{d\theta_i^k(t)}{dt} = \sigma_i^k(t) - \theta_i^k(t), \tag{6}$$

the Potts network additionally expresses latching dynamics, the key to its possible role in short-term memory.

The unit-specific thresholds $\theta_i^A$ and $\theta_i^B$ describe local inhibition, which in the cortex is relayed by at least 3 main classes of inhibitory interneurons [21] acting on $GABA_A$ and $GABA_B$ receptors, with widely different time courses, from very short to very long. In the Potts network it has proved convenient, in order to separate time scales, to consider either very slow or very fast inhibition [19, 22]. Here, we consider a more realistic case in which *both* slow and fast inhibition are taken into account. Formally, we have two inhibitory thresholds $\theta_i^A$ and $\theta_i^B$ (to denote fast, $GABA_A$ and slow, $GABA_B$ inhibition, respectively) that vary in the following way:

$$\tau_A \frac{d\theta_i^A(t)}{dt} = \gamma_A \sum_{k=1}^{S} \sigma_i^k(t) - \theta_i^A(t), \tag{7}$$

$$\tau_B \frac{d\theta_i^B(t)}{dt} = (1 - \gamma_A) \sum_{k=1}^{S} \sigma_i^k(t) - \theta_i^B(t), \tag{8}$$

where one sets $\tau_A < \tau_1 \ll \tau_2 \ll \tau_B$ and the parameter $\gamma_A$ sets the balance of fast and slow inhibition. If $\gamma_A = 0$, we have only slow inhibition in the network. If $\gamma_A = 1$, we have only fast inhibition. We have both for $0 < \gamma_A < 1$. In this way, we make a small step towards a realistic network, while maintaining relative mathematical simplicity and the ability to apply a separation of time scales to better understand the phenomenology.

We define an order parameter called *overlap*, which measures the distance between the network state and each pattern.

$$m^\mu(t) \equiv \frac{1}{Na(1-a/S)} \sum_{i=1}^{N} \sum_{k=1}^{S} \left( \delta_{\xi_i^\mu, k} - \frac{a}{S} \right) \sigma_i^k. \tag{9}$$

The overlap $m^\mu$ is normalised in such a way that it takes the value of 1 when the network state is fully aligned with one pattern.

With adaptation, the Potts network has four different phases of operation in the $w - \gamma_A$ phase space (Fig 1B). The first one is the trivial *no latching* phase, where the network operates just as an autoassociative (long-term) memory, with large storage capacity, but dynamics stop after the retrieval of the cued pattern. The Potts network undergoes a phase transition by changing one of the network parameters (e.g., the local feedback $w$ in Fig 1B). Above a phase transition, the network spontaneously latches, i.e., it generates a sequence of items, clearly defined but limited in length in the *finite latching* phase, and indefinite but progressively less well defined in the third phase, the *infinite latching* one, in which latching dynamics go on indefinitely after the initial associative retrieval. In the fourth phase the retrieved pattern is not destabilised by adaptation, and remains as a steady state. We call this the *stable attractor* phase.

As the network hops from memory to memory, it can simulate free recall. This happens if latches are concentrated onto STM items, but otherwise free, i.e., not coerced by external agents. Key to such latching dynamics is that the specific thresholds $\theta_i^k$'s inactivate, when

rising, only the corresponding attractor state and not the cortical patch *tout court*, allowing for a large variety of ensuing trajectories.

## Incorporating short-term memory function

The Potts network has so far been studied as a model of long-term memory, but it can be tweaked in minimal ways to serve also as a model of short-term or working memory. While it remains a simple object to study, it demonstrates how memory operating on widely different time scales can utilise the very same neural representations and the same associative mechanisms, based on plausible and *unsupervised* synaptic plasticity rules.

The core idea is that a few memory items, or sequences of items, are strengthened by increasing the value of some pre-existing parameter (Fig 2A). The increase, which cannot be presumed to be precisely determined, should be in any case moderate, to effectively bring only those items across a network phase transition, into a phase in which they or their sequences are held in short-term memory, effectively separate from the ocean of all items and all possible sequences in long-term memory (Fig 2B). So it is just an extra boost, without adding new components. The increase or extra boost is assumed to be temporary, and once it subsides, the short-memory has vanished. A critical assumption is that, since whatever plasticity in the

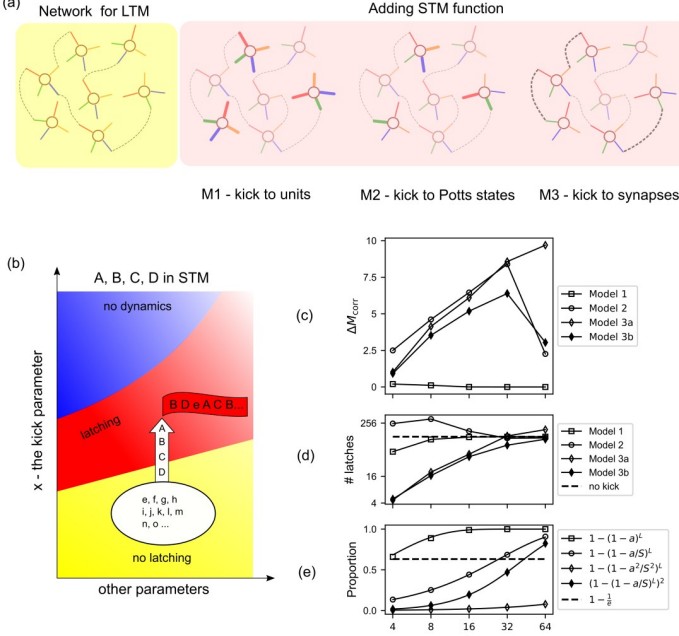

**Fig 2. Different models for holding items in STM yield qualitatively different recall performance.** (a): Schematic of the way STM is implemented in the three models. Model 1 acts at the unit-level, Model 2 at the Potts-state level, and Model 3 at the synapse level. (b): Schematic diagram of models for STM. The STM function is produced by a "boost" $\Delta x$ in the parameter $x$, representing $w$, $\theta$ and $J$ for Model 1, 2, and 3, respectively. (c): The quantity $\Delta M_{corr}$ has a maximum at around $L \simeq 32$ for Model 2 and 3b and it continues to grow for Model 3a, while it remains always close to zero for Model 1. The abscissa is $L$, the number of items in STM, in log scale. The ordinate is $\Delta M_{corr} \equiv M_{corr}(\Delta x = 0.3) - M_{corr}(\Delta x = 0.0)$, where $M_{corr}$ is the number of recalled STM items until the network either repeats an already-visited item or (mistakenly) retrieves one of the LTM items. (d): The different propensity to latch, i.e., to make transitions, is quantified by the number of latches per sequence, plotted as a function of $L$ for the 3 models, in a log-log scale. The strength of the boost is, again, $\Delta x = 0.3$ for each model. The horizontal dashed line indicates the number of latches per sequence when all $p$ patterns are on equal footing, i.e., there is no boost. (e): The proportion of resources utilised in the models predicts the peak of the performance $\Delta M_{corr}$. The dashed horizontal line indicates the proportion equal to $1 - \frac{1}{e}$. Across all 3 panels, parameters are $p = 200$, $S = 7$, $a = 0.25$, $\gamma_A = 0.5$ and $w = 1.1$.

brain serves as the extra boost, it has a transient time course, we should model it by modifying parameters in simple and coarse ways, in contrast with what we assume to happen when encoding long-term memories, which in principle can be refined over many repetitions/recall instances, and can be taken therefore to reflect very precisely set parameters, down to the level of individual synaptic efficacies.

Different neural-level mechanisms can constrain latching dynamics to a small subset of activity patterns that represent items in long-term memory. It can be envisaged that several of them may operate in synergy. Here we analyse three, which can be simply associated with distinct parameters of the Potts network, and we consider each mechanism separately from the other two, to demonstrate its characteristics ([Fig 2A]). The parameters we focus on are the degree of local feedback (Model 1), the local adaptive thresholds (Model 2) and the strength of long range connections (Model 3). In each case, a single parameter is therefore varied across many network elements, so that $L$ patterns, those supposed to be held in short-term memory, are driven into the latching regime ([Fig 2B]). This change, which embodies short-term *storage*, should avoid pushing into the latching regime also the other $p - L$ patterns, but to some extent their involvement is unavoidable, as will be shown.

**Model 1: Stronger local feedback for the items held in STM.** The first mechanism models increased depth of the attractors in the patches of cortex where any of the $L$ patterns is active, which could reflect a generic short-term potentiation of the synaptic connections among pyramidal cells in those patches, what in the Potts network is summarily represented by the parameter $w$ ([10, 19]). In the model, each of the $L$ items is active over $aN$ Potts units, and their active states are shared with many other items not intended to go into STM. This is the coarseness that leads to limited capacity of memory: if $L$ is too large, virtually all of the units are given the boost, all with the same strength, and no distinction between the $L$ selected items and the other $p - L$ remains. Formally, instead of common $w$ for all Potts units, we introduce

$$w_i = w + \Delta w \, \Theta \left( \sum_{\mu=1}^{L} \sum_{k=1}^{S} \delta_{\xi_i^\mu, k} \right), \tag{10}$$

where $\xi_i^\mu$ is the state of pattern $\xi^\mu$ at the unit $i$, $\Theta(\cdot)$ is the Heaviside step function and $\delta_{\xi_i^\mu, k}$ is the Kronecker's delta symbol.

If a unit participates in the representation of any one of the $L$ patterns in STM, then $w_i = w + \Delta w$. If not, $w_i = w$.

**Model 2: Lower adaptive threshold for the items held in STM.** In the second mechanism, a parameter regulating firing rate adaptation is reduced selectively for the neurons that are active, in those patches, in the representation of the $L$ items. That is, we *decrease* adaptation, by subtracting from the adapted threshold ($\theta_i^k$) a term $\Delta\theta$, for the Potts states that are active in any one of the $L$ patterns,

$$\tau_2 \frac{d\theta_i^k(t)}{dt} = \sigma_i^k(t) - \theta_i^k(t) - \Delta\theta \, \Theta \left( \sum_{\mu=1}^{L} \delta_{\xi_i^\mu, k} \right). \tag{11}$$

**Model 3: Stronger long-range connections for the items held in STM.** The third mechanism we consider is the one acting on the long-distance synaptic connections between neurons, represented in the Potts network [10] by the tensor connections between Potts units. We model short-term potentiation of the synaptic connections by stronger tensor connections. Since the latter connect separate Potts units, however, in order to specify exactly which tensor

elements are considered to be potentiated, we have to specify whether the $L$ patterns, in the task, are taken to be stored simultaneously. We consider two opposite cases. If they are assumed to be all stored at separate times, the stronger tensor elements are those that connect Potts states of two units both active in any one of the $L$ patterns. If they are assumed to be all stored in STM together, the stronger elements are all those that connect Potts states of two units both active in any *pair* of the $L$ patterns. We call them variants $a$ and $b$ of Model 3.

**Model 3a: Model 3 with only autoassociative connections in short-term memory.**

$$\tilde{J}_{ij}^{kl} = J_{ij}^{kl} + \Delta J \, \Theta \left( \sum_{\mu=1}^{L} \delta_{\xi_i^\mu, k} \delta_{\xi_j^\mu, l} \right), \tag{12}$$

where $J_{ij}^{kl}$ is the strength of connections that do not belong to any one of $L$ patterns in STM, given in Eq (1). Here we say that a connection belongs to a pattern when the two states that are paired by the connection participate in the representation of the pattern.

**Model 3b: Model 3 with all associative connections among STM items.**

$$\tilde{J}_{ij}^{kl} = J_{ij}^{kl} + \Delta J \, \Theta \left( \sum_{\mu=1, \nu=1}^{L} \delta_{\xi_i^\mu, k} \delta_{\xi_j^\nu, l} \right), \tag{13}$$

where $J_{ij}^{kl}$ is again given in Eq (1). In this model, we potentiate extra connections in addition to those that are potentiated in Model 3a. These are the so-called heteroassociative connections that connect Potts states of one item to those of another item in STM.

**Different models for holding items in STM are differentially effective.**   For the sake of a fair comparison among the mechanisms (Models 1, 2 and 3), we equalize the values of all parameters as they affect the $L$ patterns, so that in practice, rather than bringing them into the latching regime, which is what should happen in the real process, in our model evaluation we push the other $p - L$ out, or partially out, in different directions.

We first consider how effective are the three mechanisms in constraining latching dynamics to the $L$ items in STM. We find that for Models 2 and 3a, latching dynamics are effectively constrained to the $L$ items, but only up to a given value of $L$ (see Fig 2C, where we have shown the result for specific values of the parameters, e.g. $\Delta x = 0.3$, but those are representative of a broad range, as shown in S1–S4 Figs). The effectiveness is measured, in Fig 2C, by a quantity called $M_{\text{corr}}$, which is the number of recalled STM items until the network either repeats one of already-visited items or retrieves one of the LTM items. We then consider the difference between this quantity and the value it would have without any differentiation between the $L$ and the other items, $\Delta M_{\text{corr}} \equiv M_{\text{corr}}(\Delta x = 0.3) - M_{\text{corr}}(\Delta x = 0.0)$; this subtraction of the chance level quantifies the genuine effect of $\Delta x$. Here $x$ represents $w$, $\theta$ and $J$ for the 3 models, respectively. When we increase $L$, there are two main factors that affect $M_{\text{corr}}$. The first one is the exploration by the trajectory, resembling that of a random walk, which increases $M_{\text{corr}}$. Due to this effect $M_{\text{corr}}$ should grow like $\sqrt{L}$ as a function of $L$ (see S1 Appendix) if there are no *errors*, i.e. recall of items that are not in short-term memory. The occurrence of errors is the second factor that affects $M_{\text{corr}}$, progressively more as $L$ increases. When $L$ is small, the first factor dominates and as a result, $M_{\text{corr}}$ grows. Beyond a certain value of $L$, there is an avalanche of errors as there are many LTM patterns that are kicked as strongly as those in STM. This avalanche of errors causes the sudden drop of $\Delta M_{\text{corr}}$ seen for Model 2 and 3a in Fig 2C. We can attempt to understand this limitation as being due to interference from the LTM items, that start to dominate the dynamics at different values of the list size $L$. To illustrate this, let us consider the proportion of elements (units, states and connections for Model 1, 2 and 3, respectively) that are enhanced for a given number $L$. If we randomly pick, respectively, one unit,

state or connection, then the probability of it belonging to one of $L$ patterns in STM can be written, respectively, for Models 1, 2, 3a and 3b:

$$M1 : P_L = 1 - (1-a)^L \tag{14}$$

$$M2 : P_L = 1 - \left(1 - \frac{a}{S}\right)^L \tag{15}$$

$$M3a : P_L = 1 - \left(1 - \frac{a^2}{S^2}\right)^L \tag{16}$$

$$M3b : P_L = \left(1 - \left(1 - \frac{a}{S}\right)^L\right)^2 \tag{17}$$

All of these quantities approach 1 when $L$ becomes very large, as all elements become used towards encoding the list in STM. As a rough estimation, we can set a somewhat arbitrary criterion of $P_L = 1 - \frac{1}{e}$, above which more than half of all elements are used, and the network cannot easily discriminate STMs from LTMs. We can then roughly estimate the "critical" value of $L$, $L_c$, at which $P_L$ reaches this criterion, with which we obtain, using the parameters for which we run the simulations ($S = 7$, $a = 0.25$):

$$M1 : L_c = \frac{-1}{\log(1-a)} \approx 3.5 \tag{18}$$

$$M2 : L_c = \frac{-1}{\log(1-a/S)} \approx 27.5 \tag{19}$$

$$M3a : L_c = \frac{-1}{\log(1-a^2/S^2)} \approx 783.5 \tag{20}$$

$$M3b : L_c = \frac{\log(1-\sqrt{1-1/e})}{\log(1-a/S)} \approx 43.5 \tag{21}$$

The considerations above point to the different values of the critical list length $L_c$ obtained through the different models. This is to be expected as the different models act on different elements of the network. Model 1 has very limited capacity to constrain latching dynamics, in that interference effects occur already for low values of $L$. In contrast, Models 2 and 3b yield broadly similar values, whereas Model 3a, acting on the long-range connections, is not affected by interference until much higher $L$ values. This is because in this case, the boost is affecting a subset of the very many $NCS(S-1)/2$ tensor connection values (Fig 2E). Note that increasing the strength of the "boost" does not affect the critical list length $L_c$ (S1–S4 Figs).

However, the different manipulations intended to add short-term functionality to the network also affect its regime of operation, such that its ability to spontaneously recall, or latch, is altered, affecting the length of the sequences uttered by the network [19, 23]. The Potts network becomes able to serve STM functions once it undergoes a phase transition from the no-latching phase to the finite-latching phase. For this reason we are also interested in the propensity to latch expressed by each of our models. To investigate this propensity to latch, we first cue the network with one of the memorised patterns, after which we count the total number of

transitions that occur until the dynamics stop on their own (Fig 2D). We can see that with Model 2, constraining the dynamics to be among the $L$ items actually *enhances* the length of the sequences, whereas the opposite is true, at least up to moderate values of $L$, for Model 3 (and incidentally, for Model 1). This is because for Model 2, the direct manipulation of the adaptive threshold $\theta_i^k$ screens its "refractory" effect, affecting also sequence length. The same does not hold for Model 3, in which the adaptive threshold is not manipulated. We deduce that two aspects of Model 2 are relevant as a model of short term memory. First, the "coarseness" of Model 2 yields a limit to the list size that can be effectively enhanced. Second, the basic propensity to latch also falls off with increasing list size, reminiscent of the slowing down of retrieval from memory as the set size increases [4]. Note that the representation of objects has been found to be "enhanced" in working memory tasks [24], likely with higher neural activity in the participating units [25], broadly consistent with Model 2. Therefore, in the remainder of this work, we focus on Model 2.

## Results

### Can "free recall" by the Potts network model experimental data?

Having discussed three different models for short-term recall, we study in detail Model 2, and focus now on a specific paradigm, free recall. In free recall, participants are given a list of items to remember, and are then immediately asked to recall the items, in the order they wish. Experimental data from decades ago show that the number of items recalled from memory obeys a power law of the list length [8, 26]. To explain this finding and more generally to investigate the putative mechanisms that could hinder recall, a theoretical model for memory recall has been proposed. We refer to this model as the SAM++ model, as it was developed by Sandro Romani, Misha Tsodyks and colleagues [12, 27], with some roots in the SAM theory of Raaijmakers and Shiffrin [28], which however does not envisage the deterministic loops that terminate the search dynamics in SAM++ model. In this model, $L$ STM items are drawn from a virtually unlimited reservoir of (LTM) memory items. Transitions are defined to occur deterministically between items that have the largest similarity; as a consequence, recall trajectories always enter a loop, at which point old items are repeatedly recalled, and no new items are recalled beyond the number $R$ reached with those in the loop. Given such simple transition rules, the power-law dependence $R \propto \sqrt{L}$ can be derived (a similar derivation can be found in S1 Appendix). In a more recent study, this power law dependence has been observed for lists of up to 512 words [12].

**If limited by repetitions, the network *can* recall up to $\sim \sqrt{L}$ STM items.** In contrast to the SAM++ model mentioned above, the dynamics in the Potts network model are not deterministic (we will discuss this point in the subsection on free recall, below), and we hardly ever observe a loop in the network trajectories; hence we cannot apply quite the same stopping criterion to determine how many items have been recalled in a simulation. However we can still compute a measure somewhat similar to $R$, labeled as $M_{it}$, as the number of retrieved patterns until the network repeats one transition—which would be the first element in a loop, given deterministic dynamics. Compared to $\ln R \propto 0.5 \ln L$ (see [12]), $M_{it}$ has a steeper scaling with $L$, but still sublinear (Fig 3A). Alternatively, we can look at the number $M_{i1}$ of retrieved items until the network simply revisits one of those already visited. In contrast to $M_{it}$, $M_{i1}$ grows now *less* than a square root of $L$ (Fig 3A). To get at an intermediate behaviour, we could then define a third measure $M_i$, as the number of recalled items until one item is repeated *twice*. This somewhat contrived quantity has a behaviour indeed similar to that theoretically expected from the quantity $R(L)$, that is, a slope of 0.5 in a log-log plot (Fig 3B).

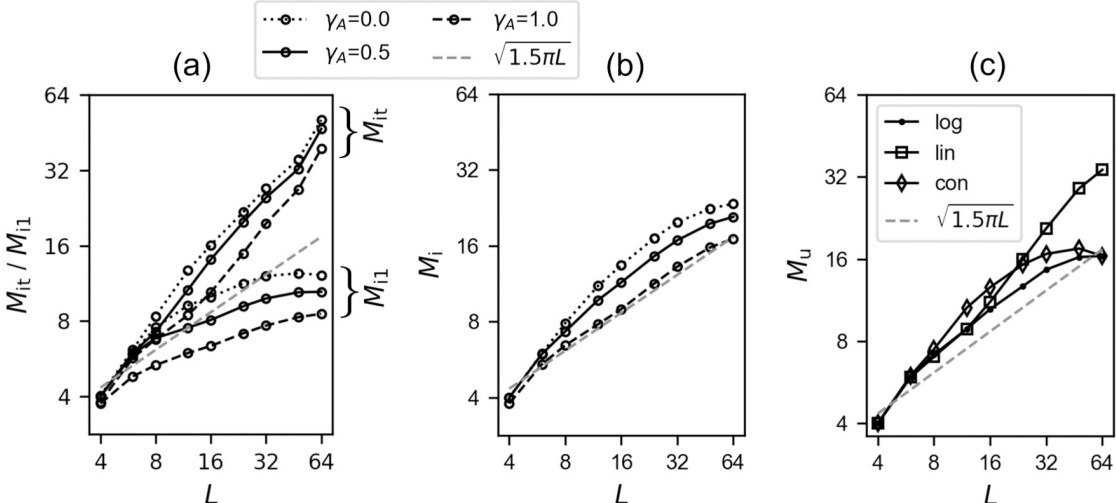

**Fig 3. Whether limited by repetitions or in duration, Potts free recall approaches a $\sqrt{L}$ dependence.** The dashed gray line is the theoretical prediction of $R$ in [12]. Both axes are in a log scale. (a): $M_{it}$ is the number of recalled STM items until one transition is repeated. $M_{i1}$ is the number of recalled STM items until one of the visited STM items is revisited. Dotted curves are for slow inhibition ($\gamma_A = 0.0$), dashed curves for fast inhibition ($\gamma_A = 1.0$), and solid ones for the intermediate regime ($\gamma_A = 0.5$). (b): $M_i$, the number of recalled STM items until one of them is repeated *twice*. In contrast to the two measures plotted in (a), this quantity approaches a square root dependence with $L$. (c): $M_u$, the number of recalled STM items within a given number of latches, $g(L)$, is plotted as a function of $L$ in log-log scale. We consider three different functions for $g(L)$: logarithmic, linear and constant, denoted by dots, squares and diamonds, respectively, for $\gamma_A = 0.5$.

In computing these three measures, we have ignored errors (extra-list items) in order to compare with [12, 27]. Note that errors are not discussed in [12, 27], in which retrieval of extra-list words is simply dismissed as irrelevant. The beauty of their treatment, in fact, stems from the simple question they pose, without getting into how the recall process happens dynamically in the brain and how LTMs affect free recall performance. These questions are those we address here, however.

Moreover, we see that whether we consider only very slow or only very fast inhibition, as in previous analytical studies [19, 22], or a more plausible balance of the two, the network behaves similarly in terms of short-term memory function. Based on this observation, hereafter we only concentrate on the balanced, or intermediate regime ($\gamma_A = 0.5$).

**If limited by duration, the network *can* again recall up to $\sim \sqrt{L}$ STM items.** In the free recall experiment conducted in [12], they computed $R$ as the number of correctly recalled words (or sentences), ignoring errors and repetitions. The time allocated to recall started from 1 minute and 30 seconds for $L = 4$, and was increased by the same amount when the length of the list was doubled. As it is problematic to establish a correspondence between human recall time and simulation time in the Potts network, we define another quantity: we compute the number of correctly retrieved items, ignoring errors and repetitions, $M_u$, within a *given number of consecutive latches*, denoted by $g(L)$. Given the stochasticity of the network dynamics in visiting pattern space, the specific choice of $g(L)$ has implications on $M_u$. We attempt to obtain a reasonable comparison with the results in [12] by writing $g(L) = 4\log_2(L) - 2$. We find that this measure has a slope of approximately 0.5 (Fig 3C). However, if $g(L) = L$, i.e., a linear function of $L$, $M_u$ has a higher slope. Finally, if we set $g(L)$ to $g(L_{max}) = 22$, with $L_{max} \equiv 64$, i.e. constant and equal to the maximum number of latches in the logarithmic option, then $M_u$ becomes slightly larger for intermediate values of $L$, suggestive of a drop after hitting a maximum. This again indicates that the Potts network can capture the empirical trend of $\sqrt{L}$, provided one

adopts a suitable rule for limiting the length of latching sequences. Of course, in experiments limiting the time available to subjects imposes implicit limits also on the errors and repetitions they can make.

**Free recall of nodes on a 2D grid also shows a $\sim \sqrt{L}$ dependence.** That the various $M$ measures obey quasi-square-root functions of $L$ may be partially understood by considering a random walk in pattern space, with equally probable visits to each of the patterns (S5 Fig) [29, 30]. Inspired by this observation, we have designed simple experiments in which subjects are asked to remember a random trajectory on a 2-dimensional grid (Fig 4A). We then asked participants to freely recall the positions of the presented dots by clicking on their positions on the grid.

Clearly, the parameters of the experimental protocol can be expected to affect recall, including the amount of time allocated for recall. However, in our experiment, participants only need to click on the correct locations (as opposed to typing in the words they recall [12]), and setting a fixed recall time may seem *ad hoc*. As an alternative, and to further explore the validity of latching dynamics as a model for this experiment, we give participants a limited number of clicks per trial, set as $2L - h(t|L)$, where $h(t|L)$ is the number of correctly recalled dots up to that point in time. Then we compute $M_R$, defined as the number of correctly recalled dots for a given $L$ ignoring errors and repetitions, and compute the same measure from simulations with the Potts network (see Methods for a description of the experiment).

We find a reasonable agreement between the performance of the Potts network and human subjects in our experiment, where both show a slope of approximately 0.5 (Fig 4B). This suggests that latching dynamics capture some aspects of the underlying neural mechanisms of free memory recall, related to the random walk nature of the trajectory, although the exact details depend on the paradigm.

(a) (b)

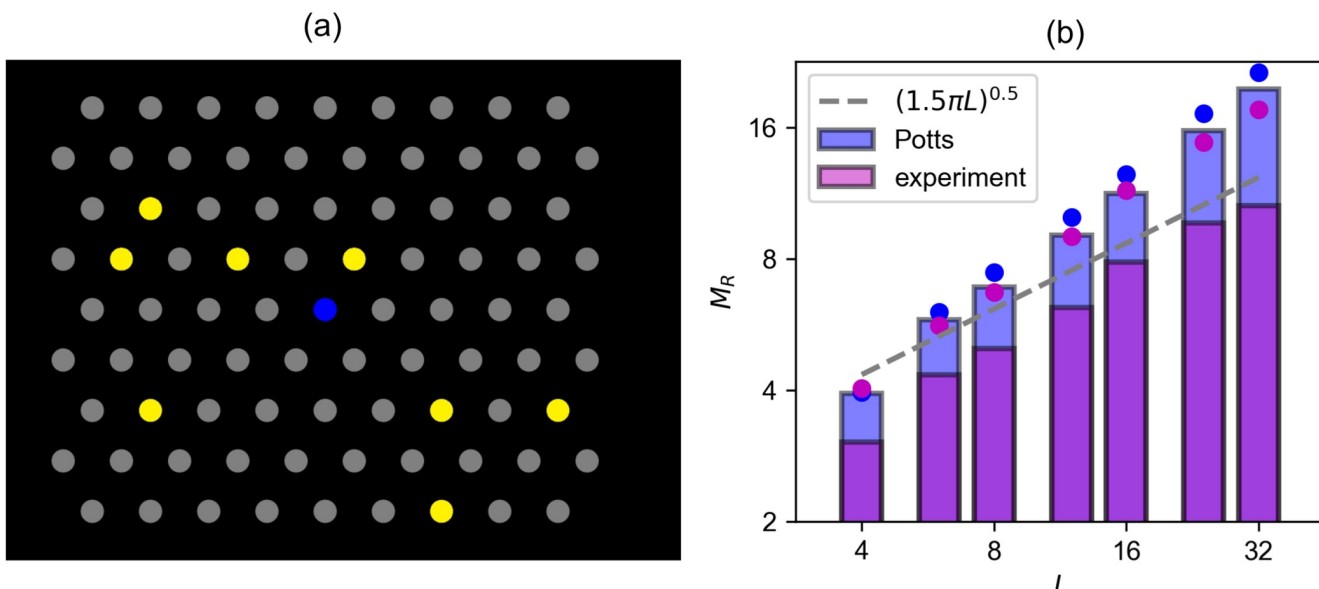

**Fig 4. Free recall of locations in a 2D grid also shows an approximate $\sqrt{L}$ dependence.** (a): The 2D grid used in the free recall experiment. Yellow dots show one example of stimuli with $L = 8$. (b): $M_R$, the average number of correctly recalled locations in our experiment, is shown by the height of pink bars in a log-log scale. The distance from the bar to the dot of the same colour corresponds to the standard deviation of the mean. Results of 40 participants are pooled together. The same quantity $M_R$ is computed, from simulating Model 2, as the number of correctly retrieved STM items within a given number of consecutive latches set as $2L - h(t|L)$, where $h(t|L)$ is the number of correctly recalled STM items up to that point in time (blue bars). The dashed gray line is the theoretical prediction of $R$ in [12]. Both results, from our experiment and the Potts network, show an approximate $\sqrt{L}$ trend.

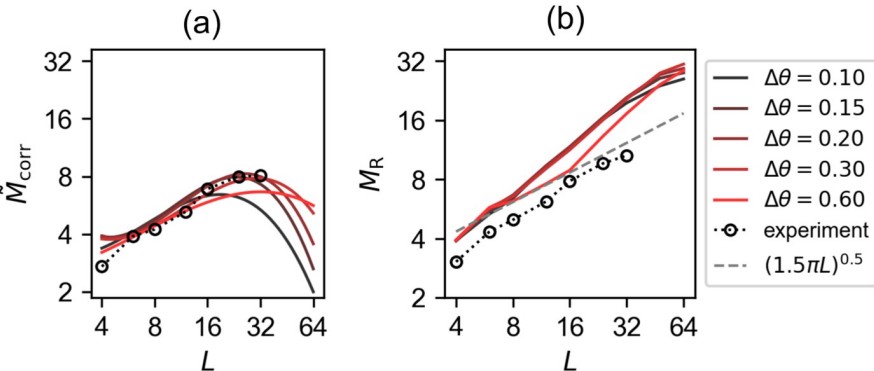

**Fig 5. An error-limited measure of recall has a maximum value.** Two measures, $\tilde{M}_{\mathrm{corr}}$ and $M_{\mathrm{R}}$, are shown for several values of $\Delta\theta$, coded by colours. Black dotted curves are the experimental results of free recall of locations in a 2-dimensional grid. (a): $\tilde{M}_{\mathrm{corr}}$ has a maximum value. It is the number of recalled STM items until the network either revisits one of the already-recalled STM items or visits one of the LTM items, but within a given number of latches $-2L - h(t|L)$, where $h(t|L)$ is the number of correctly recalled STM items up to that point in time. (b): $M_{\mathrm{R}}$ shows a scaling behaviour. $M_{\mathrm{R}}$ is the number of recalled STM items, ignoring repetitions and errors, within a given number of consecutive latches, again $2L - h(t|L)$.

**If limited by errors, the network cannot recall beyond its STM capacity.** The measure $M_{\mathrm{corr}}$ was introduced and discussed above to compare three different models. Here we compute the same quantity with a slight modification; in order to compare with our experimental data, we consider sequences of variable length that depends both on list length $L$ and time. We consider again lengths $g(L) = 2L - h(t|L)$, where $h(t|L)$ is the number of correct STM items already retrieved; within this sequence we count the number of correctly retrieved STM items up to the first error or repetition. We compute this quantity $\tilde{M}_{\mathrm{corr}}$ for several values of $\Delta\theta$ in the Potts network. The behaviour of $\tilde{M}_{\mathrm{corr}}$ with respect to $L$ is qualitatively similar to that of the experimental curve for a broad range of $\Delta\theta$ values (see Fig 5A). For all values of $\Delta\theta$, $\tilde{M}_{\mathrm{corr}}$ saturates reaching a maximum that is similar to that of the experimental data, of around 8 items correctly recalled. Exceptions are at the two extremes: too small and too large values lead to lower capacity of the Potts model, below 7 items.

The saturation behaviour, and hence the notion of memory capacity, again contrasts with the scaling behaviour approximated by the various measures such as $M_{\mathrm{i}}$, $M_{\mathrm{u}}$ and $M_{\mathrm{R}}$. This contrast holds irrespective of the values of network parameters used in simulations. Indeed the scaling behaviour of $M_{\mathrm{R}}$ is almost independent on the value of $\Delta\theta$ except when it is too large, $\Delta\theta = 0.6$ (Fig 5B). Furthermore, we find that the two contrasting behaviours—scaling and saturation—are fairly robust to change of network parameters such as $\Delta\theta$, $S$ and $a$ (S6 and S7 Figs).

"Performance" therefore depends very differently on $L$, if recall is taken to be terminated by errors, i.e. by the erroneous recall of an item that is not in STM. Thus, while if ignoring errors the notion of STM *capacity* appears irrelevant (given the scaling behaviour of the various quantities discussed above), it becomes quite relevant if errors are considered to be critical in the task.

In summary, we have shown that whether we get scaling or saturation in STM performance depends on the specific metric we use to measure it, both in the Potts network, endowed with an STM mechanism and in our experiment. In free recall experiments, performance has often been quantified through the $M_{\mathrm{R}}$ index, thereby ignoring errors. This scaling behaviour appears to hold even up to 512 items [12]. In contrast, taking our experiment as an example, we have shown that if errors are considered critical, in our case through the $M_{\mathrm{corr}}$ measure, then the

performance of human subjects actually expresses a saturation at about 8 items. In our model, that expresses a similar behaviour, this saturation is brought about by the interference from long-term memories.

## Serial recall

Can the Potts model endowed with short term memory express also behaviour similar to *serial recall*? This is a paradigm very similar to free recall, but with a crucial difference. Here, participants are instructed to recall items in *the same order* as they have been presented, making the task more difficult and, for a model, to rely on random walk dynamics would appear to be counterproductive. Clearly, the network model requires some extra ingredient to produce ordered sequences.

First, in light of the literature pointing at how STM span depends on the nature of items being remembered [14, 15, 31, 32], we have performed serial recall experiments with three different types of items, but within the same general paradigm. We asked participants to observe and repeat sequences of stimuli presented to them on the screen—either digits or spatial locations on a 2-dimensional grid (Fig 6A), and varied the time of presentation of the stimuli in the observed sequence. There were two conditions for the spatial locations, referred to as Locations and Trajectories: in the Locations condition, considered to involve only "discrete" items, the six chosen locations around the centre of the grid were highlighted in any order, while in the Trajectories condition, every next location was one of the six consecutive locations around the previous one, thus suggesting a "continuous" trajectory. Contrary to the free recall experiment reported above, in this task participants had to recall the material in the correct order, otherwise the trial was dismissed as incorrect. Participants started with short sequences of length 3; if they recalled them correctly in at least 3 out of 5 trials, the sequence length increased, until a memory capacity limit for this stimulus type and presentation time was reached. Fig 6B shows the capacity for serial recall in this task (see Methods for how we computed the memory capacity).

(a)                                                          (b)

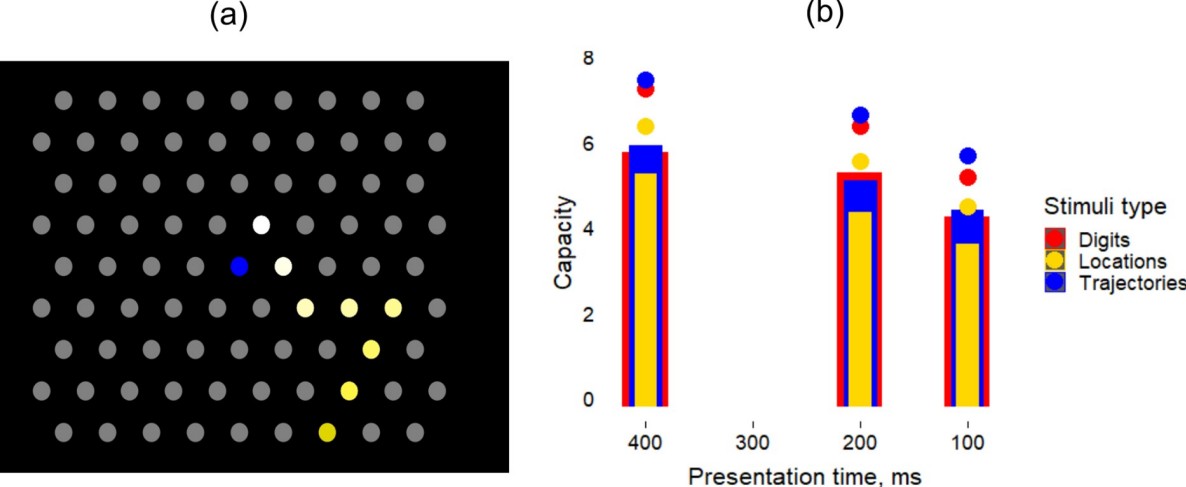

**Fig 6. Short-term memory capacity for serial recall does not markedly depend on stimulus type.** (a): The 2D grid used in the serial recall experiment. Dots are presented sequentially as shown by the highlighted dots here ($L = 8$). (b): Memory capacity for serially presented stimuli for different presentation times: bars correspond to the average capacity across participants, while the distance from the bar to the dot of the same colour corresponds to the standard deviation of the mean. We performed the experiment for three different stimulus types, shown in different colours.

Our experiment yields two main results (Fig 6B). The first is that the type of stimulus does not affect the recall probability, except for a slight disadvantage in the *discrete* Locations condition, suggesting a universal mechanism for recall independent of the material, which manifests itself at the systems level. The second, which is more pronounced, is the effect of presentation time per stimulus, that, when shortened, makes it more difficult to correctly remember and repeat the longer sequences, suggesting a disadvantage at the *encoding* stage. We ask whether latching dynamics in the Potts model can reproduce this finding. Given that our results, as well as those from other studies [4], show limited dependence on stimulus material, hereafter we only consider the result with digits in order to establish a comparison with our model.

We used Model 2 (lower adaptive threshold for items held in STM) to constrain the dynamics into a subset of $L = 6$ patterns intended as the 6 digits of our experiment. In addition to that, we introduced *heteroassociative* weights, similar to Model 3, to provide the sequential order of presented digits (see Eq (24) in Methods).

We find a good agreement between our experimental data and the model (Fig 7). In addition, we find that human subjects perform better if the to-be-memorised digit series include ABA or AA (Fig 7A and 7C), in line with the notion that the repetition of an item aids memory [33–36]. Such sequences are not produced by our model, due to firing rate adaptation and inhibition preventing the network from falling back onto the same network state for time scales of the order $\tau_2$.

The heteroassociative component of the learning rule (Eq (24) in Methods) provides "instructions" to the network regarding the sequential order of recall, allowing it to perform serial recall (this is to be contrasted with the model with a purely autoassociative learning rule, performing free recall). The strength of such instructions is expressed through the parameter $\lambda$. We find that this parameter plays a role similar to that of presentation time in our experiments; increasing it enhances performance, just as increasing the presentation time increases the performance of human subjects (Fig 7). However, values of $\lambda$ that are too large again make performance worse and deteriorate the quality of latching (Fig 7E). The dynamics becomes a stereotyped sequences of patterns, see S8 Fig, without really converging towards attractors, and the sequence itself is progressively harder to decode. Therefore, the most functional scenario is when the heteroassociative instruction acts as a bias or a perturbation to the spontaneous latching dynamics rather than enforcing strictly guided latching in the Potts model. This is in sharp contrast with the mechanism for sequential retrieval envisaged in the model considered in [37], where the heteroassociative connections are the main and only factor driving the sequential dynamics; in that case, without it, there are no dynamics but rather, at most, the retrieval of only the first item. The effect of lower adaptive threshold (expressed by $\Delta\theta$) on latching sequences is to constrain the dynamics to a subset of presented items among $p$ patterns, but values of $\Delta\theta$ that are too high degrade the performance as well as the quality of latching (Fig 7B, 7D and 7E).

As mentioned above, the Potts model produces latching sequences even without any heteroassociative instructions. This means that the free transition dynamics of the model may or may not coincide with the "instructions" provided by the heteroassociative weights. Then one question naturally arises. How does the congruity between spontaneous, endogenous sequences and instructed ones affect the performance of the model? To see this effect, we obtain some intrinsic latching sequences by running simulations with $\lambda = 0$; from these latching sequences, we generate a set of instructions for the serial order. These instructions are *congruous*, inasmuch as they reproduce latching sequences emerging without any heteroassociative instructions. Then we compare the performance for these congruous instructions with those of incongruous instructions, which we obtain by shuffling the congruous

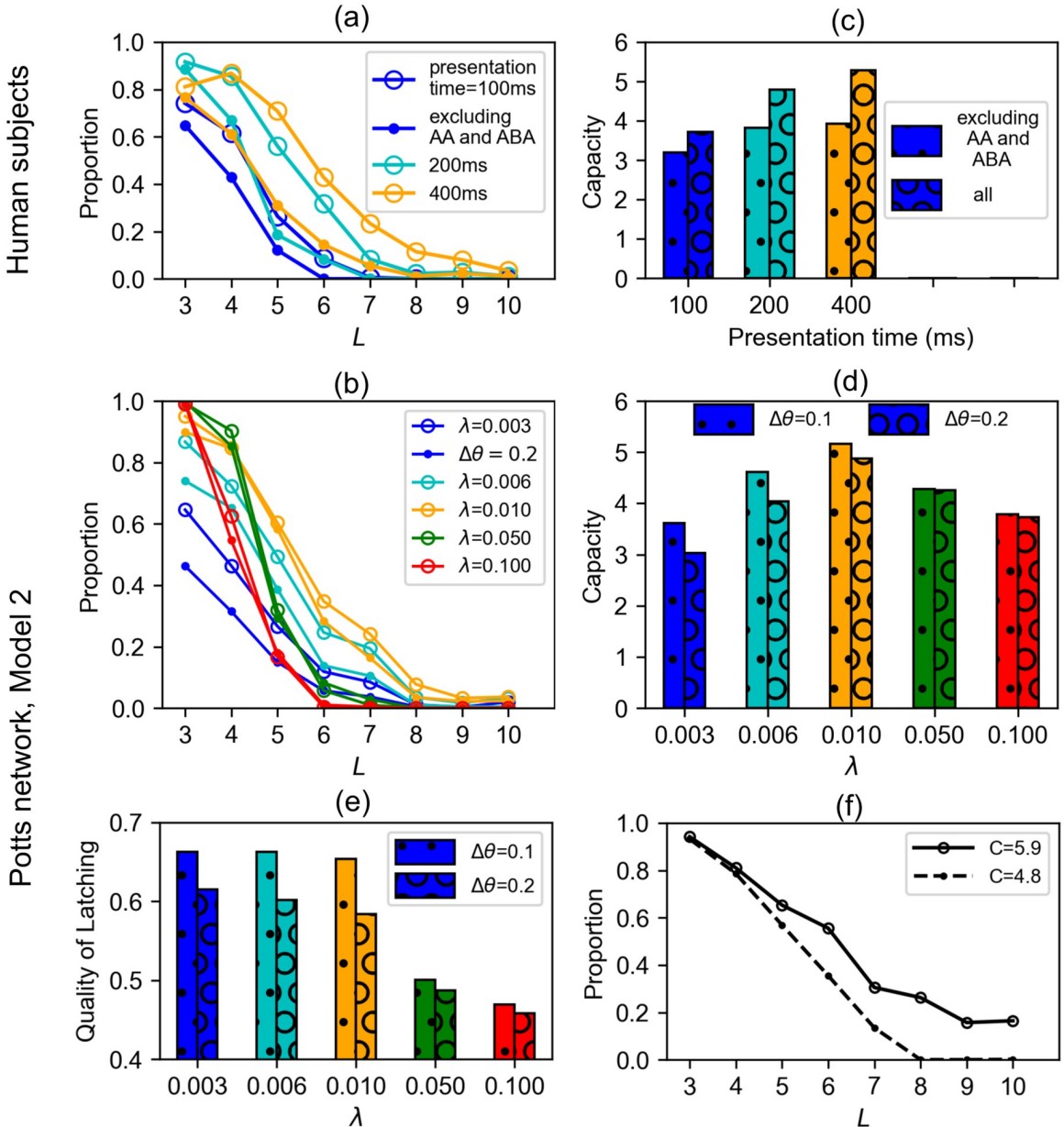

**Fig 7. Serial recall of digits by human subjects and the Potts model.** (a): Proportion of correct trials in the serial recall task with digits. Data for all subjects ($n = 36$) are pooled together. Colour codes for presentation time (in ms). Dots are for sequences without repetitions like AA and ABA and circles are for all sequences. (b): Proportion of correct subsequences in a latching sequence of the Potts model. Colour codes for values of the heteroassociative strength $\lambda$, that hard-codes transitions into the weights. Circled (dotted) curves correspond to simulations with the boost $\Delta\theta = 0.1$ (0.2). (c): Memory capacity computed from the curves of (a), (see Methods). (d): Recall capacity computed from latching sequences of the Potts model is shown by the same colour-coding as in (b). (e): The quality of latching (see Eq (26)), a measure of the discriminability of the individual memories composing a sequence, is shown for different values of $\lambda$ and $\Delta\theta$. (f): Proportion of correct subsequences in a latching sequence of the Potts model for $\Delta\theta = 0.1$, $\lambda = 0.01$. The solid curve is for congruent instructions only and the dashed curve is for a shuffled version of intrinsic sequences.

ones. We find that the capacity of the model increases by as much as 1 item for the congruous case relative to the incongruous case (see the legend in Fig 7F).

These results together with those from the previous two sections indicate that intrinsic latching dynamics, similar to a random walk, can serve short-term memory (e.g., they can be

utilised by free recall). Furthermore latching dynamics can also serve serial recall, if supplemented by biases that modify the random walk trajectory; the modification (or perturbation) should be a quantitative one, which biases the random walk character of the trajectories, rather than an all-or-none, or qualitative one, that inhibits it. This is consistent with our recent experimental result, to be reported elsewhere, where "guided" serial recall leads to poorer performance than a non-guided control.

### The trajectories in free recall

In previous sections we saw a reasonable agreement between some experimental measures and those extracted from simulating the Potts model. This agreement essentially results from two factors: first, the Potts model can produce a sequence of discrete activity patterns even though its governing equations are continuous at the microscopic level; and second, the dynamics of the Potts model visit the patterns in a random-walk like process. We now examine the sequences more closely to see what factors influence latching sequences and how the network wanders around the landscape of memorized patterns.

We first ask ourselves: once the network is cued with a given pattern, what elicits the retrieval of the next one? In previous studies [19, 22], it was shown that transitions occur most frequently between highly correlated patterns, when the Potts model serves a long-term memory function. We confirmed that this is also the case when the Potts model serves a short-term memory function, as in the current study (S9 Fig). Indeed, the larger the average correlation of one pattern with all other patterns in STM, the more often it is visited by the network (S10 Fig). This result is consistent with a recent experimental study on how memorability of words affects their retrieval in a paired-associates verbal memory task [38].

Next we probe the flow of information in the latching sequences of the STM model embedded in the Potts neural network by computing the normalised mutual information between two patterns as a function of their relative separation in a latching sequence, $z$ (see Methods for details). We find that the mutual information is decreasing rapidly with respect to $z$, with a quasi-periodic modulation, reminiscent of the temporal profile of intensity of a damped oscillator (Fig 8A). The periodic modulation is much more evident for $L = 16$ than for $L = 64$; within the range of $z$ we have considered, we see a peak at $z \approx 4.5$ for $\gamma_A = 0.0$ and at $z \approx 3.5$ for $\gamma_A = 0.5$, but we also see the second peak at $z = 6$ in addition to the first peak at $z = 3$ for $\gamma_A = 1.0$ (Fig 8A). The second peaks for $\gamma_A = 0.0$ and $\gamma_A = 0.5$ would be located at $z \approx 9$ and $z \approx 7$, respectively. The quasi-period of the "damped oscillation", $\zeta$, is twice the $z$–value of the first peak, therefore, decreasing with increasing $\gamma_A$, starting from $\zeta \approx 9$ at $\gamma_A = 0.0$ until $\zeta \approx 6$ at $\gamma_A = 1.0$. For $L = 64$, it is as if the damping ratio is too high to observe any periodicity.

This behaviour is related to how the Potts network "freely" forages the landscape of the embedded attractors. We visualize this nontrivial behaviour for $\gamma_A = 0.5$, where we not only see a kind of damped *wave* that "propagates" along the $y$–axis with the variable $z$ as an effective "time", but also see the "reflection" of the wave around $z \approx 3.5$ (Fig 8C).

What causes these characteristics of the latching trajectories of the Potts model? To answer this question, we define a quantity, called $d$, which is an index of "semantic" distance between two patterns in their representational space. We defined a distance between two patterns $\mu$ and $\nu$ as follows.

$$d(\mu, \nu) \equiv \frac{C_{ad}(\mu, \nu) - C_{as}(\mu, \nu) + 1}{2}, \tag{22}$$

where $C_{as}$ and $C_{ad}$ measure the correlation between two patterns (see Eqs (27) and (28) in Methods).

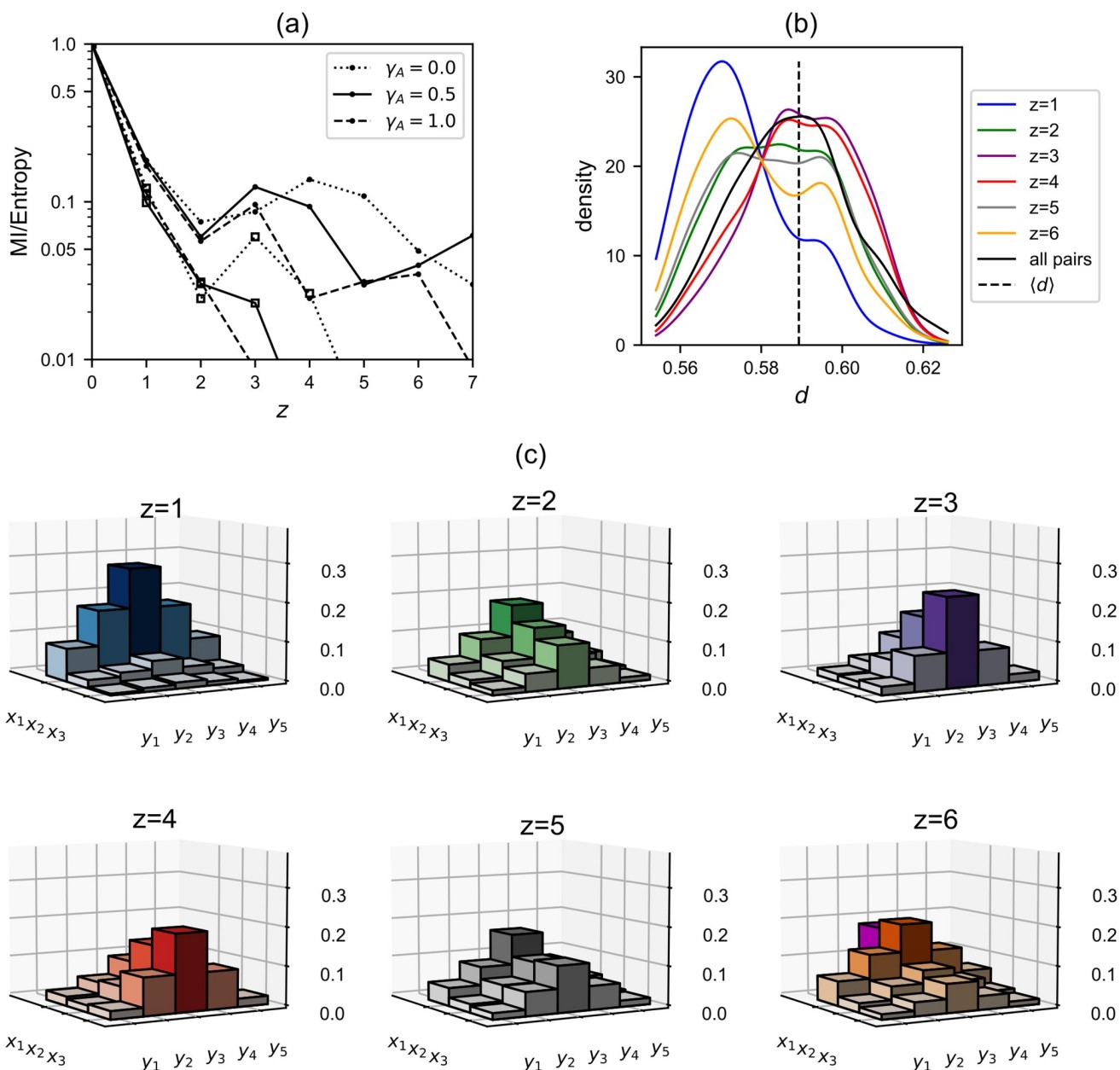

**Fig 8. Damped waves in pattern space.** (a): Mutual information as a function of the relative separation of two patterns in a latching sequence, $z$. The ordinate is the mutual information $I(z) \equiv I(\mu, \nu|z)$ (see Methods for details) divided by the entropy $H$. Note the logarithmic scale of the $y$–axis. Parameters are $\Delta\theta = 0.3$, $L = 16$ (64) for the curves marked with dots (open squares), $w = (0.4, 0.8, 1.0)$ for $\gamma_A = (0.0, 0.5, 1.0)$. (b): Distribution of distance, $d$, between two patterns that have the relative separation $z$ in a latching sequence for $L = 16$, $\gamma_A = 0.5$ and $w = 0.8$. The black, vertical line indicates the mean value of $d$ across all $p$ patterns. The solid black curve is the PDF of $d$ among all possible pairs between $L$ patterns in STM. (c): Histograms for the visiting frequency of patterns in STM, given one pattern is recalled. The remaining $L - 1 = 15$ patterns are arranged along the $x$–axis by their visiting frequency at the next position of the currently retrieved pattern in a sequence ($z = 1$), giving three groups $x_1$, $x_2$ and $x_3$ of 5 patterns each. Each group is further arranged symmetrically along the $y$–axis, with the most frequent pattern on the midline ($y_3$). Visiting frequency is double-encoded by the height and colour of bars. The lonely, magenta bar behind the group $x_1$ shows the visiting frequency of the currently recalled pattern once it returns at the position $z$.

We consider the distribution of $d(\mu_n, \mu_{n+z})$, the distance between two patterns that are separated by $z$ latches in a latching sequence, for 6 values of $z$ (Fig 8B). At $z = 1$, latching occurs mostly between highly correlated patterns as expected, where the higher correlation is expressed by lower $d$. At the second step in a latching sequence ($z = 2$), patterns that have

higher $d$ values than the average value $\langle d \rangle = \frac{S-2}{2S} a + \frac{1}{2} \approx 0.589$ show a comparable proportion of the probability density curve relative to patterns with lower values of $d$. Then the proportion of higher $d$ values is much larger than the proportion of lower $d$ values for $z = 3$ and $z = 4$. This means that the network prefers to visit those patterns that are less correlated with the initially retrieved one at the third and fourth step. So we can say that the network reaches the most "distant" pattern from its "initial" pattern around $z = 3.5$, which is the "reflection" point of the wave (Fig 8C). As $z$ increases further to reach 6, the density curve is getting closer to the curve for $z = 1$, thus approaching the periodicity mentioned above. This periodicity is confirmed by S11 and S12 Figs.

These results indicate that latching trajectories by Potts networks have a quasi-random walk character, though biased by correlations between patterns in their representational space. This is consistent with earlier applications of latching dynamics to semantic priming [13].

## Discussion

The Potts model offers a plausible cortical framework to discuss aspects of memory dynamics, without losing too much of the clarity afforded by simpler non-neural models. Indeed, a major difficulty with network models of memory storage in the human cortex, which have attempted to reflect its dual local and long-range connectivity [17, 39] by articulating interactions at both the local and global levels, is that their mathematical or even computational tractability usually has required *ad hoc* assumptions about memory organization. For example, the partition of memory items in a number of classes, in each of which memories are expressed by the activity of the same cortical modules [40]—which makes it awkward to use such a network model to analyse the free or serial recall of arbitrary items. On the other hand, more abstract models have provided brilliant insight [27] which is hard, however, to relate to neural variables and neural constraints. By subsuming the local level into the dynamics of individual Potts variables, the statistical analysis can focus on the cortical level, what is effectively a reasonable compromise.

The (global) cortical level is in particular the one to consider in assessing short-term memory phenomena, in which interference from widely distributed long-term memories plays a central role. Experiments with lists of unrelated words are a prime example [13]. The free energy landscape of the Potts model provides a setting for quasi-discrete sequences of states, with properties that turn out to be similar to those of random walks. This happens, however, only within a specific parameter range, and only to a partial extent, so that often one has in practice several intertwined sequences, with simultaneous activation of multiple patterns, as well as pathological transitions, all characteristics with potential to account for psychological phenomena, and which are lost in a more abstract purely symbolic model. We have thus discussed three generic neural mechanisms that may contribute to restrict the random walk, approximately, from $p$ to $L$ items. Although not exclusive, we have argued that the second such mechanism is the one most relevant to account for the recall of list of unrelated items.

To model the recall of ordered lists, an additional *heteroassociative* mechanism can be activated, which biases the random walk, but again approximately, resulting in frequent errors and limited span. We have observed that, at least in the Potts network, if the heteroassociations, which amount to specific instructions, dominate the dynamics, the random character is lost. With it we lose the entire latching dynamics—which cannot be harnessed to just passively follow instructions.

In summary, a Potts network can generate quasi-discrete sequences from analog processes, with the possibility of errors in

1. the "digitalisation" into a string of discrete states, one at a time

2. the restriction to $L$ out of $p$ item in LTM

3. the order, both in the specific sense of serial order, and in the generic one of avoiding repetitions.

These possibilities for error reflect weaknesses of latching dynamics as a mechanism for short-term memory expressed by a Potts network, and at the same time underscore the value of the mechanistic model, inasmuch as similar "flaws" crop up in the phenomenology. The analysis of such flaws can lead to refinements of the model.

Thus, point 2, the difficulty of restricting latching dynamics to a subset of all the long-term memory representations, is made even more severe in paradigms that involve multiple subsets. For example, in analyses of the Phonological Output Buffer (POB) the hypothesis has been considered of mutiple POBs, one holding simple phonemes, one function words, one numerals, etc., conceptually as sort of separate drawers, or *mini-stores* [41]. If one accepts the evidence of a common substrate for working memory and long-term memory representations [42], one cannot resort to different "drawers", i.e., different scratchpads or the like, where to temporarily hold the items from distinct subsets, and this makes enforcing the restriction more difficult. Likewise, one cannot regulate the correlation between the long-term representations, as one could do if new *ad hoc* representations were temporarily set up. These constraints can result in intrusions, a simple form of false memory, e.g. by items that are strongly semantically associated to items in a short-term memory list [43], or by items in prior lists [44]. It would be tempting to pursue a fully quantitative study of these phenomena [45] to try and extract constraints, for example, on the time course of the "boost" that models STM in the Potts network.

In relation to point 3, latching dynamics are intrinsically stochastic in nature, even in the absence of microscopic noise, because of the heterogeneity of the underlying microscopic states. With randomly correlated representations, trajectories among items are effectively random, with only a tendency to avoid close repetitions, as a result of the adaptation-based mechanism. Interestingly, a tendency to perceive random processes as less prone to repetition than they really are is a hallmark of human cognition [46]. Beyond the vanilla version of the model, however, it is rather trivial to incorporate e.g. adjustments of the time course of the boost, to produce primacy and recency, or adjustments of the correlations between pairs of representation to produce preferred transitions. What is more interesting and still lacking, to our knowledge, is again a quantitative study of the degree of randomness of the recall process, in the context of remembering lists for example—a study made inherently difficult by the need to use novel items in a within subjects design. The same need effectively prevents the analysis of the recalled string at the single neuron level: even when recording the activity of neurons in awake patients, only generic forms of selectivity can be reliably studied, e.g., that expressed by putative "time" cells [47]. Interestingly such a study has been recently carried out in rats, pointing at the random walk character of the spatial trajectories they recall shortly after experiencing them [29]. While a similar approach cannot easily be extended to humans, to probe the dynamics of individual neurons, the Potts model can help interpret evidence at the integrated cortical level.

It is its fallibility in the production of a simple string of items, however, where the Potts network offers crucial insight beyond that provided by simpler and more abstract models, in which the digitalisation of a string is *a priori* given. Latching dynamics can involve partially parallel strings, items incompletely recalled simultaneously with others, periods of utter confusion, stomping attempts. Statistically, they are all observed with prevalence determined by the

various parameters. These flaws in the analog-to-digital transduction of the Potts model may be useful in the interpretation of electrophysiological data. One basic question in this domain is: can two items be simultaneously active in working memory? On this question, experimental evidence has been difficult to obtain, because a process that appears to involve two items active together, might in fact rapidly alternate between them. Recently, however, the genuinely concurrent activation of two items has been reported with a model-based analysis of EEG data [48]. In that study, holding on to the two items meant better performance in the task, so it reflects a capability, not a flaw of the short-term mechanism. If extended to sequences of endogenously generated states, as the Potts model indicates would occur, at least in certain regimes, it would mean that not only the focus of attention when performing a similar task need not be unique, but also that parallel streams of thoughts can be entertained along partially interacting trajectories. This could be applied to interpret electrophysiological measures of mind wandering dynamics [49], with significant implications for our intuition about the unity of consciousness [50].

## Methods

We studied the latching dynamics of the Potts network by extensive computer simulations. In a simulation the network is first initialized by setting all variables at their equilibrium values. Then we cue the network with one of the memorized patterns, remove the cue and let the dynamics proceed. Simulations are terminated if the network shuts down into a globally stable null attractor (in which all units are inactive) or if the total number of updates reaches $10^5$.

### The Potts network as a model for short-term recall

The Potts network has been studied so far as a model of long-term memory; but it can also serve, with minimal modifications, short-term or working memory. It suffices to strengthen a few memory items, or sequences of items, by increasing the value of some pre-existing parameter, to effectively bring the network across a phase transition, as indicated in Fig 2. Evidence and arguments supporting the model of short-term memory as an activated portion of long-term memory can be found in [7].

The types of modifications we consider, in this study, all implement the assumption that, when a subject is performing a task of immediate recall, the attractors corresponding to the presented items have been facilitated at the encoding stage. We can visualize them as becoming wider and deeper in their basins. At the recall phase, then, we interpret that an item has been recalled by the Potts network if its activity becomes, at least for a brief time, most correlated with the corresponding attractor, among all LTM items. The facilitation of attractors for STM items can be done by changing distinct parameters of the network. We propose in Models three different models for short-term memory function.

### The Potts model for serial recall

We use Model 2 to approximately constrain the dynamics to a subset of $L_0$ patterns, for example the 6 digits of our experiment. We have $p = 200$ patterns in long-term memory, among which we give a $\Delta\theta$ boost to $L_0 = 6$ patterns, indicated as 1, 2, 3, 4, 5, 6. In addition to the auto-associative connections between Potts units given by Eq (1), we introduce heteroassociative connections to mimic the sequential order of the items presented in the experiment; we randomly pick $L$ items among the 6 items (1, 2, 3, 4, 5, 6), allowing repetitions. When $L = 6$, for example, it can be $2 \rightarrow 4 \rightarrow 3 \rightarrow 2 \rightarrow 5 \rightarrow 1$. But we do not include sequences that have a subsequence like $AA$ or $ABA$ because the Potts model cannot really express such sequences (they occasionally appear in the dynamics, but only when the transition from A to B is incomplete

or anomalous). Sequences without subsequences of the form ABA and AA are favoured by the Potts network. So, we prepare a set of 80 sequences that do not include any subsequence of the form ABA and AA, for a given value of $L$, with $L = 3, 4, \ldots, 10$. If we denote a sequence of this set as $I_1, I_2, \ldots, I_L$, then the model for serial recall is determined by the following equations

$$\tau_2 \frac{d\theta_i^k(t)}{dt} = \sigma_i^k(t) - \theta_i^k(t) - \Delta\theta\Theta\left(\sum_{\mu=1}^{L_0} \delta_{\xi_i^\mu, k}\right) \tag{23}$$

$$J_{ij}^{kl,het} = \lambda\Theta\left(\sum_{\mu=1}^{L-1} \delta_{\xi_i^{I_{\mu+1}}, k}\delta_{\xi_j^{I_\mu}, l}\right) \tag{24}$$

$$h_i^k = \sum_{j\neq i}^{N}\sum_{l=1}^{S}(J_{ij}^{kl}\sigma_j^l + J_{ij}^{kl,het}\theta_j^l) + w\left(\sigma_i^k - \frac{1}{S}\sum_{l=1}^{S}\sigma_i^l\right) \tag{25}$$

## Definition of relevant quantities

The quality of latching is evaluated by means of $d_{12} - Q$. $d_{12}$ is the difference between the largest overlap and the next largest one, averaged over time and over so called *quenched* variables [22], while

$$Q = \frac{1}{T}\int_{t_0}^{t_0+T} q(t)dt, \tag{26}$$

is the average overlap with the next $L$ patterns, since $q(t) \equiv \frac{1}{L-1}\sum_{i=1}^{L-1} m^{\mu_i}(t)$. $m^{\mu_i}$ is the overlap of the network activity with a pattern $\mu_i$ and $\mu_1, \ldots, \mu_{L-1}$ are the $L - 1$ patterns having largest overlaps excluding the maximum overlap. This quantity is a kind of measure on how "condensed", i.e., partially recalled, the non-recalled patterns are.

The correlation between patterns is measured by two quantities [18, 19],

$$C_{as}(\mu, v) = \frac{1}{Na}\sum_{i=1}^{N}(1 - \delta_{\xi_i^\mu, 0})\delta_{\xi_i^\mu, \xi_i^v}, \tag{27}$$

$$C_{ad}(\mu, v) = \frac{1}{Na}\sum_{i=1}^{N}(1 - \delta_{\xi_i^\mu, 0})(1 - \delta_{\xi_i^v, 0})(1 - \delta_{\xi_i^\mu, \xi_i^v}). \tag{28}$$

The average values of $C_{as}$ and $C_{ad}$ over different realizations of randomly-correlated patterns are given by

$$\langle C_{as}\rangle = a/S, \tag{29}$$

$$\langle C_{ad}\rangle = a(S - 1)/S. \tag{30}$$

Mutual information $I(z)$ is computed by

$$I(z) = \sum_{\mu}\sum_{v} P(\mu, v|z) \log_2 \frac{P(\mu, v|z)}{P(\mu)P(v)},$$

where $P(\mu, v|z)$ is the joint probability of observing pattern $\mu$ at the position $n$ and observing pattern $v$ at the position $n + z$ in a latching sequence, with $n$ can be any integer between 1

and the length of the latching sequence. $P(\mu)$ is the marginal probability of observing pattern $\mu$ at any position of the latching sequence. Mutual information is then normalised by entropy,

$$H = -\sum_{\mu} P(\mu) \log_2 P(\mu). \tag{31}$$

## Network parameters

The network parameters used in this study are set as in S1 Table, if not specified explicitly.

## Experiments of free recall and serial recall

Both experiments were conducted online, with participants recruited through https://www.prolific.co/.

**Serial recall.** The 36 participants were instructed to watch a sequence appear on the computer screen and repeat the sequence just after, by clicking on the screen. They had to repeat sequences of $L$ stimuli ($L$ starting from 3). In each of the conditions, they had 5 trials for each length $L$, with $L$ incremented by one until 3 out of 5 trials were incorrect; the last $L$ is then taken as the limit capacity for this participant in this condition. For each participant the sequences were of all three stimulus variants:—(D) Digits out of {1, 2, 3, 4, 5, 6} on a black screen, presented one at a time—(L) Locations on a hexagonal grid highlighted one by one, out of 6 around the central (blue) dot—(T) Trajectories on the same hexagonal grid: now each consecutively highlighted dot is one of 6 neighbors of the previous one (as shown in Fig 6A, the first one is always one of the six around the center). Each stimulus was presented for one of the durations (in separate blocks): 400ms, 200ms, 100ms. First always came the 400ms training session, then either 200ms or 100ms (balanced), and then the remaining duration. Presentation order was balanced across duration and stimulus material. In additional experiments, landmarks on the grid were used as well as intermediate presentation times, but no significant effect on the recall performance was observed.

In order to measure the memory capacity in this serial recall task, we first plot the proportion of correct trials as a function of $L$ either for each participant in Fig 6B or for the pooled data across all participants in Fig 7A. Although the minimum value $L$ we used was 3, we added two "data points" by hand to the proportion-$P(L)$, setting it to 1 (i.e., a putative 100% for $L = 1$ and $L = 2$). We then compute the memory capacity as the simple sum,

$$C = \sum_{L=1}^{L_{\max}} P(L),$$

where $L_{\max}$ is the maximum value of $L$ used in the experiment. This measure is usually referred to as *Area Under the Curve* or AUC [51].

**Free recall.** The same hexagonal grid as in serial recall is used (Fig 4A). In this experiment, the sets of stimuli were presented all at once, and the participants ($N = 40$) were instructed to repeat as many as they could recall, by clicking on the dots in the grid. For each set size $L$ in {4, 6, 8, 12, 16, 24, 32}, the participants had 5 trials to do, each trial allowing for $2L$—(number of correctly recalled items) clicks. For example, if participants correctly clicked 3 correct dots out of 4 times in a trial with $L = 4$, they had another chance, to reach the fourth correct dot, as $2L - 3 = 5$. A set of size $L$ was presented for $\log_2(L)$ seconds.

## Supporting information

**S1 Fig. $\Delta M_{\text{corr}}$ is shown for several values of $\Delta w$ from simulating Model 1.** The abscissa is the number of items in STM, $L$, in a log scale. The ordinate is $\Delta M_{corr} \equiv M_{\text{corr}}(\Delta w) - M_{\text{corr}}(0)$, where $M_{\text{corr}}$ is the number of recalled STM items until the network either repeats an already-visited item or (mistakenly) retrieves one of the LTM items. Left: $w = 1.0$, right: $w = 1.1$.
(PDF)

**S2 Fig. $\Delta M_{\text{corr}}$ is shown for several values of $\Delta\theta$ from simulating Model 2.** Details as in S1 Fig.
(PDF)

**S3 Fig. $\Delta M_{\text{corr}}$ is shown for various several values of $\Delta J$ from simulating Model 3a.** Details as in S1 Fig.
(PDF)

**S4 Fig. $\Delta M_{\text{corr}}$ is shown for various several values of $\Delta J$ from simulating Model 3b.** Details as in S1 Fig.
(PDF)

**S5 Fig. The quantity $R$ is shown as a function of $L$, obtained from simulations with SAM++ model ([12, 27]) and with the Potts network endowed with long-term memory function.** This quantity ($R$), which is the number of visited STM items until the search process enters a loop, is well-defined only in the case of symmetric similarity matrix. In other cases the quantity $R$ is ill-defined; a closed loop is hardly ever observed in search process, so we compute, instead, $M_{\text{i1}}$, which is the number of visited STM items until the network revisits one of the already-visited items, as a surrogate for $R$. The blue curve with squares is $R(L)$ obtained from simulating SAM++ model with random symmetric similarity matrices (1000 simulations). The blue curve with circles is $R(L)$ obtained from simulating SAM++ model with random non-symmetric similarity matrices (10000 simulations). In both cases elements of similarity matrices are drawn from a uniform distribution between 0 and 1. In the latter case, the degree of symmetry is 0.5 on average. The green line with diamonds is $R(L)$ obtained from simulations of the Potts model without short-term boost in the intermediate inhibition regime ($\gamma_A = 0.5$, $w = 1.4$). We randomly pick $L$ out of $p = 200$ patterns and treat them as if they were STM items. The solid black line is from the numerical evaluation of Eq (1) in S1 Appendix, which is derived from an equal-probability assumption. All lines shown here have a slope of approximately 0.5.
(PDF)

**S6 Fig. Behaviour of $M_{\text{i}}(L)$ is fairly robust to the values of $\Delta\theta$.** $M_{\text{i}}(L)$ is plotted for several values of $\Delta\theta$ from simulating Model 2. $M_{\text{i}}$ is the number of recalled STM items until one of them is repeated *twice*.
(PDF)

**S7 Fig. $M_{\text{i}}$, $M_{\text{corr}}$ and $M_{\text{R}}$ remain qualitatively the same with respect to changes in $S$ and $a$.** These quantities remain qualitatively the same with respect to changes in $S$ and $a$, as long as latching dynamics are stably maintained under these changes. $M_{\text{corr}}$ is the number of recalled STM items until the network either revisits one of the already-recalled STM items or visits one of the LTM items, but within a given number of latches $- 2(L - h(t|L))$, where $h(t|L)$ is the number of correctly recalled STM items up to that point in time. $M_{\text{R}}$ is the number of correctly retrieved STM items within a given number of consecutive latches set as $2(L - h(t|L))$, ignoring errors and repetitions.
(PDF)

**S8 Fig. Too high values of λ lead to faltering latching dynamics.** In serial recall by the Potts model, too high values of λ, relative strength of heteroassociative connections to the autoassociative ones, lead to faltering latching dynamics. Two example sequences are shown, for the same parameter values: $\omega = 1.0$, $\gamma_A = 0.5$, $\Delta\theta = 0.1$, $\lambda = 0.05$. Each colour corresponds to a different pattern. The proportion of simulation in which latching completely fails, as in the right panel, increases with λ.
(PDF)

**S9 Fig. Scatter plot with $C_{as}$ and $C_{ad}$ on the two axes.** See Eqs (27) and (28) in Methods for definitions of $C_{as}$ and $C_{ad}$. Each data point (obtained from Model 2 for $L = 64$) indicates, for enhanced clarity, an average over 3 pairs of patterns. Crosses (open circles) represent correlations averaged over 3 most (least) frequent pairs, whose relative positions are determined by $z$ in a latching sequence. Horizontal and vertical dashed lines indicate the average values of $C_{as}$ and $C_{ad}$ over all patterns. At the first step ($z = 1$), latching occurs most frequently between highly correlated patterns, in agreement with previous studies on long-term memory. At the third step, the trend is reversed.
(PDF)

**S10 Fig. Patterns that are visited more frequently seem to be those that share a larger number of active units with a larger set of patterns, reflected in the correlation matrix.** (a): Re-ordered transition matrix for $p = 200$ and $L = 16$ for one set of patterns, ordered according to the visit frequencies of each pattern in that data set. The matrix of transition probability has rows—where the network latches from, which in turn is just the probability of appearance of each pattern—that look roughly similar to the average row (with fluctuations), while the columns—where the network latches to—are very different from each other, from the heavy ones on the left to the light ones on the right. (b): $C_{as}$ matrix (see Eq (27) in Methods for its definition), again ordered in the same way as in (a). The diagonal has been set to 0 artificially, in order for off-diagonal values to be more visible. (c): Mean correlation of each pattern in STM with all the others in STM, $y_n$, versus its visit frequency $f_n$ for $p = 200$ and $L = 16$. Numbers indicate the pattern indices (16 of them).
(PDF)

**S11 Fig. Probability density of $d(\mu_n, \mu_{n+z})$ shows the quasi-periodic evolution with respect to $z$.** Probability density of $d(\mu_n, \mu_{n+z})$ (see Eq (22)) is divided by the probability density of $d(\mu, \nu)$ for all possible pairs among $L$ patterns in STM from simulating Model 2. From $z = 1$ to $z = 6$, we can see the quasi-periodic evolution of the probability density function. Parameters are $w = 0.8$, $\gamma_A = 0.5$, $L = 16$, $\Delta\theta = 0.3$.
(PDF)

**S12 Fig. Visiting frequency of a pattern at the position $n + z$ as a function of $d(\mu_n, \mu_{n+z})$ and $d(\mu_{n+1}, \mu_{n+z})$ from simulating Model 2.** Colour indicates the visiting frequency. From the upper left panel to the lower right one, we can see that the brightest spot (most frequent visits) rotates counter-clockwise. Dashed black lines indicate the average value across all pairs in STM on the corresponding axis. Parameters are $w = 0.8$, $\gamma_A = 0.5$, $L = 16$, $\Delta\theta = 0.3$.
(PDF)

**S13 Fig. Mutual information is plotted up to $z = 9$ for confirming the peoriodicity stated in Fig 8.**
(PDF)

**S1 Appendix. Deriving scaling law under the assumption of equal visits.**
(PDF)

**S1 Table. Parameter values used in simulations.**
(PDF)

## Acknowledgments

We acknowledge intensive discussions with both HFSP and M-GATE collaborations, and in particular with the Prof. Misha Tsodyks and Dr. Mikhail Katkov, Michelangelo Naim.

## Author Contributions

**Conceptualization:** Kwang Il Ryom, Vezha Boboeva, Alessandro Treves.

**Data curation:** Oleksandra Soldatkina.

**Formal analysis:** Vezha Boboeva.

**Funding acquisition:** Alessandro Treves.

**Investigation:** Kwang Il Ryom, Oleksandra Soldatkina.

**Software:** Kwang Il Ryom.

**Supervision:** Alessandro Treves.

**Writing – original draft:** Kwang Il Ryom.

**Writing – review & editing:** Vezha Boboeva, Oleksandra Soldatkina, Alessandro Treves.

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
