## [Editor Report · Decision Letter 0]

8 Mar 2021

Dear Mr. Ryom (and Allesandero),

Good news and bad news. The good news is that I really liked the explanation you provided of why our short term memory is so bad. The bad news is that, in my opinion, the paper was written in a way that made it next to impossible to figure out what was going on. I think I did figure it out, but mainly because I had an idea before starting. Without that, I would have been completely lost. And, given that my understanding of the paper was based mainly on priors, I may be totally wrong. So my main comment is: completely rewrite the paper, with a semi-naive reader in mind. I personally try to write my papers so that a math-competent person with no background in the particular subject can follow it relatively easily.

I'll give some specific comments below, but they're only to provide a flavor; I stopped reading after a while, as I got more and more lost. I believe you'll be instructed to provide a point-by-point rebuttal to these comments. You don't have to do that. Instead, all I ask is that the paper is easy for me to read. When that's the case, I'll send it out to reviewers. In its current state it's not ready.

Specific comments:

1. Probably the most serious problem is that you never actually say what's going on. Which, if I understand things, is as follows:

Start with an attractor network with p fixed points (memories). Add adaptation, so that the fixed points become unstable after a while, and the systems hops from one to the other. Then pick L fixed points to be stored in short term memory, and strengthen them. Initialize the network at one of those L fixed points, and let it wander from one fixed point to another. When the network wanders outside the sort term memory -- to one of the other p-L fixed point -- short term memory ends.

I will admit that I'm guessing on most of that. I didn't read the whole paper, but none of that was stated in the parts I did read (intro, the beginning of results and the beginning of methods). The only hints I had were on lines 11-23, which is the only description I could find of latching dynamics, "... recall exploits the natural tendency of the cortex to hop from state to state -- latching dynamics", and on lines 55-56, where you say "We show that by adding a mechanism that gives an extra "kick" to a small subset of L among p patterns in long-term memory, ...", which I take to mean that some memories were strengthened.

It needs to be clear to the reader, from the beginning, what's going on -- stated simply, so that one doesn't have to work through the math, or rely on priors, to figure it out. I personally would drop the phrase "latching dynamics" altogether, since it doesn't have any intrinsic meaning to me (and it conjures up all sorts of phenomena that have nothing to do with memory). But if you do use it, be sure to define it carefully, and prominently.

2. The second most serious problem is that the equations are out of order. It was not possible for me to make sense of Eqs. 1, 2 or 3, since I didn't know the underlying network equations. I had to go to Methods to make sense of them. Since they're needed to understand the paper, the network equations should go first, in the main text.

3. Several comments on the network equations (now in methods):

a. Please tell us what you use for xi_i^mu.

b. I didn't understand the definition of a. Is it a function of the xi_i^mu?

c. Why are you using the Potts model rather than a Hopfield network?

d. Presumably the theta's are what causes the network to adapt. You should provide intuition as to why.

e. What's the significance of the superscript 0?

4. I did not understand lines 124-7.

5. Line 132: what's special about 0.3?

6. Paragraph starting on line 137: the expressions for P_L should be derived. I may have been able to work them out myself, but I was pretty lost by this point, and it didn't help that I didn't know what a was.

7. Paragraph starting on line 146: you talk about sqrt{L} scaling, but isn't that ruled out by Fig. 1c, which shows that the number of latches is independent of L?

8. Fig. 2: what's gamma_A? It was probably defined somewhere, but it was not easily accessible.

9. Line 226: what's a "consecutive latch"?

At this point I was pretty lost, and stopped reading. Again, these are only to provide a flavor of the things that confused me. You should be able to make the paper easily understandable by a math-competent reader.

Peter Latham, Associate editor.

--formal letter follows

Thank you very much for submitting your manuscript "Latching dynamics as a basis for short-term recall" for consideration at PLOS Computational Biology.

As with all papers reviewed by the journal, your manuscript was reviewed by members of the editorial board and by several independent reviewers. In light of the reviews (below this email), we would like to invite the resubmission of a significantly-revised version that takes into account the reviewers' comments.

We cannot make any decision about publication until we have seen the revised manuscript and your response to the reviewers' comments. Your revised manuscript is also likely to be sent to reviewers for further evaluation.

Sincerely,

Peter E. Latham

Associate Editor

PLOS Computational Biology

Lyle Graham

Deputy Editor

PLOS Computational Biology
---

## [Decision Letter · Decision Letter 1]

14 Jun 2021

Dear Mr. Ryom,

Although this a major revision, overall the reviewers were positive, and it doesn't look like it will be _too_ hard to make the changes. Says the person who doesn't have to make them. ;)

Peter

--formal letter follows.

Thank you very much for submitting your manuscript "Latching dynamics as a basis for short-term recall" for consideration at PLOS Computational Biology.

As with all papers reviewed by the journal, your manuscript was reviewed by members of the editorial board and by several independent reviewers. In light of the reviews (below this email), we would like to invite the resubmission of a significantly-revised version that takes into account the reviewers' comments.

We cannot make any decision about publication until we have seen the revised manuscript and your response to the reviewers' comments. Your revised manuscript is also likely to be sent to reviewers for further evaluation.

Sincerely,

Peter E. Latham

Associate Editor

PLOS Computational Biology

Lyle Graham

Deputy Editor

PLOS Computational Biology

Reviewer's Responses to Questions

**Comments to the Authors:**

Reviewer #1: Authors discuss retrieval capacity limitations using different measures and in different variation of Potts model for memory storage. In general, our knowledge of memory recall mechanisms in humans is very limited. Therefore, theoretical models, especially supported by experiments may help us constraint possible memory mechanisms. Current work presents some of the results in vague form that would better be sharpened before publication. For example, in lines 339-349 it appears that authors implicitly assume that the difference between long-term memory and short-term memory is in the way one measure performance, and not a separate mechanism. But this is so vague stated and buried in the rest of the text that it is not clear whether authors make this statement or not. Authors are trying to advance the idea that capacity of recall is limited by items interference. Nevertheless, authors ignore the model that give estimated value of capacity close to one observed in experiments with humans. In contrast, they choose the model that has much larger expected capacity. The arguments to choose one specific model as the model for short-term memory are not convincing. Nevertheless, results of experiments are roughly consistent with simulations. Therefore, it is possible that the model describes some aspects of recall. In my opinion the manuscript should be revised to make points made clear.

Minor points.

Lns 112., 115. The same letter theta used for different variables which is confusing.

Ln. 129 “step towards plausibility” probably means towards realistic network

Lns 132-145. The statements may be more precise. (ln. 133 “perfectly retrieved”, ln. 136 “after the retrieval” what means retrieval? Perfect, above threshold on overlap?). (ln. 137 “Above phase transition” - is there a single parameter defining phase space?) (ln. 143. “but otherwise free” - ???)

Ln 214-215. “latching dynamics are effectively constrained to the L items, but only up to a given value of L” – meaning is not clear

Ln 241. “**ita** regime of operation, such that **the its** ability to spontaneously recall”

Section 3. It is not clear why one would like to have more latches that the number of stored memories. It is probably more important how many unique memories are recalled (latched).

Lns 339-349. The meaning of these two paragraphs are not clear. Do authors suggest that STS is not a brain mechanism but an artifact of quantification method? There are some modeling works in psychology that does not require STS per se to describe free and serial recall (e.g. doi:10.1037/0033-295X.114.3.539). But the rest of text is inconsistent with this statement. So, I guess, authors can make their statement more precise here.

Ln 368-369 “In this way we measure the memory capacity for serial recall by computing the Area Under the Curve (Fig. 6b).” – Could authors be more precise what exactly was done and why AUC appears here?

Ln 387 “such sequences as Potts-compatible” – that may be too strong statement, since one may be able to implement in the future Potts model that can recall arbitrary sequence. Reserving this strong term then would lead to confusion. This is not critical, for current presentation though.

Ln 409-410 “These instructions are congruous, as they reproduce latching sequences emerging without any heteroassociative instructions” – it is not clear what is mathematically meant by instructions.

Ln 415 “can serve short-term memory (e.g., free recall)” – meaning is not clear

Fig. 7 Blue dotted curve (a) 0.6+0.4+0.1 << UAC 3 in (c) “(c) Area Under the Curve (AUC) computed from the curves of (a), intended as a measure of overall performance.”

Ln 431. Cad is referred (Fig S10) before it is defined.

Ln 433 Cas is referred (Fig S11) before it is defined.

Reviewer #2: The authors present a model of free and serial recall, which embodies the idea of short-term memory as the result of “priming” a sub-set of items in the long-term memory store. The long-term memory store is implemented by using a Potts model, and three different mechanisms for priming items are introduced and studied. The model is able to reproduce experimental observations, notably the “square root law”, and provide interesting insights into possible mechanism(s) responsible for the limited capacity of short-term memory.

Recent work by Tsodyks and collaborators has shown that a random walk in “memory space” can account for subjects’ short-term memory performances to an amazingly quantitative detail. This work follows up on that thread and provides a more “microscopic” interpretation of the memory random walk. Importantly, in my opinion, it shows that the hypothesized dynamics can indeed occur in a distributed memory system; and strongly suggests that capacity limits are ultimately due to the distributed nature of the long-term store.

The results reported appear correct, as far as I can check. The paper is very well written and the figures are well chosen and informative. I have no comments.

Reviewer #3: The paper by Il Ryom et al is a great contribution for understanding the memory of sequences and short term memory of multiple objects. The authors employ their previously developed Potts network model of cortical memory processing and consider modifications of it (e.g. adaptation, synaptic plasticity etc) that allow a study of how short term memory of multiple objects as well as sequences can be formed from the stored long term memory patterns. In addition to this, they offer experimental data that allow understanding how each of these different modifications contribute to the errors (or regularities in the errors) that appear in the data. I think this work is important, as theoretical models for memory of sequences and short term memory, the capacity of such memories, and type of errors that they lead to have been paid much less attention to. This is particularly the case when compared to working memory models of e.g. objects, or models of long term memory. When it comes to sequences processing models that have been studied, more or less all rely on the the standard way of storing and retrieving sequences follow the seminal work of Sompolinksy & Kanter PRL 1986 where hetro-associative connectivities are employed, without much studying how the presence of long term stored memories influence errors in retrieval. The current paper takes this further by considering how other mechanisms such as adaptation and intrinsic latching (the ability of the network to generate quasi-random sequences of long term stored patterns) interact with this heteroassociative connectivity. The paper also provides a mechanistic model of the sqrt(L) law in free recall which is also quite novel. So, in principle, the paper is a suitable contribution to PLoS Comp Biol, though I think before publication, as I describe below, a number of changes should be done and a number of issues explained.

1. The first point that I think needs to be addressed by the author is how they pitch their work. The authors start by discussing their aim of explaining the low capacity of short term memory. For this, they cite a number of experimental results. The problem that I found confusing was that short term memory is one of those words that mean different things to different people. In the introduction and throughout the paper, the authors mention a range of experimental results from change detection experiments in visual working memory tasks to free recall tasks, some making quantitative observations, some qualitative, although all point to the direction of limited capacity (however defined in that particular task) of memory in those tasks. The model the authors describe, however, as far as I understand, is mainly about a system that generates a sequence of a certain number of objects in a given order, or quasi-randomly (e.g. in a free recall task). How, for instance, this relate to change detection memory tasks is not immediately clear.

I think the paper would benefit from a more focused introduction, and perhaps a section reviewing what in different and specific experimental setups people define as short term memory, what they use to measure capacity, and how the model described here relate to each. This I believe would be extremely useful for the audience of PLoS Comp Biol who may not all know all the different ways people define short term memory in experiments and how these different ways relate to each other.

2. In estimating L_c, the authors use the criterion P_L= 1-1/e combined with Eq. 14-17. where does this criterion come from?

3. In sec 4.2, the authors define g(L) as the given number of consecutive latches and then set this to 4 log-2(L)-2 to "establish a reasonable comparison with the results in [12]". Can you elaborate where this comes from?

4. Section 6. I think it would be useful if the authors explicitly give the expression for the mutual information between patterns. This can be defined in a number of ways, and it is only from the context that one can infer what the author mean by it. same section, z is defied as the relative separation. Relative to what?

5. Again in section 6, it seems to me that the authors make a number of guesses that are not clearly stated.

For instance in line 442, they state where the second order peak would be located but don't say how they come to this conclusion, or how what zeta is twice the z-value of the first peak.

I also think it is better to refer to the peaks as first and second peaks instead of first order and second order peaks.

6. Last sentence of section 6 "This is consistent ...". I think the damped oscillation results in this section are quite interesting and it would be appreciated if the authors elaborate on the consistency with the results of sec 13 and/or discuss other possible ways experiments can support or have supported this phenomena.

7. Regarding the damped oscillatory behaviour of sec 6: could this be simply understood by the fact that there is a competition between adaptation (making a more correlated patterns less likely to follow) and correlations (making a more correlated pattern to be follow)? If so, is it not possible to write down an effective dynamics that describes the phenomena based on these two?

throughout the paper the authors repeadetly mention the papers from Misha Tsodyks' group (refs 12 and 25). I think it would be useful if the authors give a name to this model, instead of using they and their together with the refs.

line 87: which model patchers of cortex each of which representing the state of a single patch of the cortex

line 90: one quiet state: perhaps it should be said that this ia state where no pattern is retrieved, e.g. a background firing state, not necessarily a silent or no-firing state.

line 106: current input

line 111: effective inverse temperature is a physics jargon. I would say that beta measure the level of noise in the system.

line 141: revise the sentence. I cannot parse it.

line 152-155: long and difficult sentence to parse. it would perhaps be easier to say "are strengthened by increasing the value of some pre-existing parameters e.g. .... This increase effectively brings ...". Also I am not sure being items across a "network phase transition" means.

line 241: its its

lines 491-494: "We have ..." long sentence, hard to parse and I think incomplete.

line 459: wouldn't it better to use discrete instead of digital? digital is often taken to be equivalent to binary though it is a wrong take.

line 504: "paradigms that involve multiple subset". Please elaborate what this means.

line 503: what do you mean by "the time course of the "boost"."

**Have the authors made all data and (if applicable) computational code underlying the findings in their manuscript fully available?**

Reviewer #1: None

Reviewer #2: Yes

Reviewer #3: Yes

PLOS authors have the option to publish the peer review history of their article (what does this mean?). If published, this will include your full peer review and any attached files.

Reviewer #1: No

Reviewer #2: No

Reviewer #3: **Yes: **Yasser Roudi
---

## [Decision Letter · Decision Letter 2]

3 Sep 2021

Dear Mr. Ryom,

There are actually some minor revisions, but they're pretty easy, so it's not worth going another round. Congratulations on a very nice paper!

Peter

---formal letter follows

We are pleased to inform you that your manuscript 'Latching dynamics as a basis for short-term recall' has been provisionally accepted for publication in PLOS Computational Biology.

Best regards,

Peter E. Latham

Associate Editor

PLOS Computational Biology

Lyle Graham

Deputy Editor

PLOS Computational Biology

Reviewer's Responses to Questions

**Comments to the Authors:**

Reviewer #1: Authors carefully addressed all points raised before. The exposition is substantially improved.

Minor points.

There are few typos, e.g. ln 740. .... log_2^L ....

Reviewer #3: I am pretty happy with the revised version and the answers to my comments. I only have to minor comments now.

1. In their response to my comments 2 and 3, the authors have made clarification in the response letter, but have not adjusted the text of the paper. I think they should also add them in the text.

2. My comment re line 141 of the previous version. Try:

"As the network hops from memory to memory, it can simulate free recall. This happens if latches are

concentrated onto STM items, but otherwise free, i.e., not coerced by external agents."

**Have the authors made all data and (if applicable) computational code underlying the findings in their manuscript fully available?**

Reviewer #1: None

Reviewer #3: None

PLOS authors have the option to publish the peer review history of their article (what does this mean?). If published, this will include your full peer review and any attached files.

Reviewer #1: No

Reviewer #3: No

---

## [Editor Report · Acceptance letter]

10 Sep 2021

PCOMPBIOL-D-21-00218R2 

Latching dynamics as a basis for short-term recall

Dear Dr Ryom,

I am pleased to inform you that your manuscript has been formally accepted for publication in PLOS Computational Biology. Your manuscript is now with our production department and you will be notified of the publication date in due course.

With kind regards,

Andrea Szabo
